



# Ice nucleation activity of silicates and aluminosilicates in pure water and aqueous solutions. Part 2 – Quartz and amorphous silica

Anand Kumar, Claudia Marcolli, Thomas Peter

Institute for Atmospheric and Climate Sciences, ETH Zurich, Zurich, 8092, Switzerland

*Correspondence to:* Anand Kumar (anand.kumar@env.ethz.ch)

**Abstract.** Divergent ice nucleation (IN) efficiencies of quartz, an important component of atmospheric mineral dust, have been reported in previous studies. We show here that quartz particles obtain their IN activity from milling and that quartz aged in water loses most of its IN efficiency relative to freshly milled quartz. Also, the quartz surface—much in contrast to that of feldspars—is not prone to ammonia-induced IN enhancement. In detail we investigate the influence of solutes on the

IN efficiency of various silica ($SiO_2$) particles (crystalline and amorphous) with special focus on quartz. We performed immersion freezing experiments and relate the reported contradictory behavior to the influence of milling, and to the aging time and conditions since milling. Immersion freezing with silica particles suspended in pure water or aqueous solutions of $NH_3$, $(NH_4)_2SO_4$, $NH_4HSO_4$, $Na_2SO_4$ and $NaOH$, with solute concentrations corresponding to water activities $a_w = 0.9 - 1.0$, were investigated in emulsified droplets by means of differential scanning calorimetry (DSC) and analyzed in terms of the

onset temperature of the heterogeneous freezing signal $T_{het}$ and the heterogeneously frozen water volume fraction $F_{het}$. Quartz particles, which originate from milling coarse samples, show a strong heterogeneous freezing peak in pure water with $T_{het} = 247 - 251$ K. This IN activity disappears almost completely after aging for 7 months in pure water in a glass vial. During this time quartz slowly grew by incorporating silicic acid leached from the glass vial. Conversely, the synthesized amorphous silica samples show no discernable heterogeneous freezing signal unless they were milled. This implies that defects provide

IN activity to silica surfaces, whereas the IN activity of a natural quartz surface is negligible, when it grew under near-equilibrium conditions. For suspensions containing milled quartz and the solutes $(NH_4)_2SO_4$, $NH_4HSO_4$ or $Na_2SO_4$, $T_{het}$ approximately follows $T_{het}^{\Delta a_w^{het}}(a_w)$, the heterogeneous freezing onset temperatures that obey $\Delta a_w^{het}$-criterion, i.e. $T_{het}^{\Delta a_w^{het}}(a_w) = T_{melt}(a_w + \Delta a_w^{het})$ with $\Delta a_w^{het}$ being a constant offset with respect to the ice melting point curve, similar to homogeneous IN. This water-activity-based description is expected to hold when the mineral surface is not altered by the

presence of the solutes. On the other hand, we observe a slight enhancement in $F_{het}$ in the presence of these solutes, implying that the compliance with the $\Delta a_w^{het}$-criterion does not necessarily imply constant $F_{het}$. In contrast to the sulfates, dilute solutions of $NH_3$ or $NaOH$ (molality $\geq 5 \times 10^{-4}$ mol kg$^{-1}$) reveal $T_{het}$ by $3 - 8$ K lower than $T_{het}^{\Delta a_w^{het}}(a_w)$, indicating a significant impact on the mineral surface. The lowering of $T_{het}$ of quartz suspended in dilute $NH_3$ solutions is opposite to the distinct increase in $T_{het}$ that we found in emulsion freezing experiments with aluminosilicates, namely feldspars, kaolinite, gibbsite

and micas. We ascribe this decrease of IN activity to the increased dissolution of quartz under alkaline conditions. The defects that constitute the active sites appear to be more susceptible to dissolution and therefore disappear first on a dissolving surface.





## 1 Introduction

The influence of cirrus and mixed-phase clouds on Earth's radiative budget is well recognized, yet not fully understood
(Baker, 1997; DeMott et al., 2010; Storelvmo et al., 2011). Ice formation in clouds may be initiated via homogeneous ice
nucleation (IN) below 237 K, whereas it requires an ice nucleating particle (INP) to occur heterogeneously at higher
temperatures between 237 K and 273 K (Pruppacher and Klett, 1994; Vali et al., 2015). Mineral dusts are a well-established
class of aerosol, consisting of various minerals, such as feldspars, clay minerals, micas, calcite and quartz, which exhibit
widely varying IN abilities (Murray et al., 2011; Atkinson et al., 2013; Kaufmann et al., 2016). The atmospheric relevance of
these different minerals as INPs depends on both their abundance in airborne dusts and their IN activity, which in turn may
depend on their production process and atmospheric aging.

For a long time, clay minerals have been considered the dominating IN active species amongst mineral dust particles. This is
because of their well-documented IN ability together with their high abundance in the fine particle fraction, which facilitates
long-range and high-altitude transport (Usher et al., 2003; Matsuki et al., 2005; Murray et al., 2012; Pinti et al., 2012).
However, they are IN active only at temperatures too low to explain many observed instances of cloud glaciation (Atkinson
et al., 2013). More recently, feldspars, and more specifically potassium-containing feldspars (K-feldspars) have been
suggested as the determinant species for the IN activity of airborne desert dusts (Atkinson et al., 2013). Yet, follow-up
studies have shown that not all K-feldspars exhibit the same high IN activity (Harrison et al., 2016; Kaufmann et al., 2016;
Peckhaus et al., 2016), and that microcline, the K-feldspar with the highest freezing temperatures, constitutes only a minor
fraction of collected desert dusts (Boose et al., 2016b; Kaufmann et al., 2016).

Quartz, the dominant dust component collected near source regions, is a crystalline mineral composed of silicon and oxygen
atoms in a continuous framework of $SiO_4$ tetrahedra, with each oxygen atom being shared by two tetrahedra. Therefore,
quartz has an overall chemical formula of silicon dioxide ($SiO_2$), also called silica (Götze and Möckel, 2014). Quartz is a
potentially relevant mineral dust for heterogeneous IN in the atmosphere (Field et al., 2006; Murray et al., 2012; Boose et al.,
2016b). Moreover, it is found in high proportions in atmospherically transported Saharan dust samples (Avila et al., 1997;
Caquineau et al., 1998; 2002; Alastuey et al., 2005; Kandler et al., 2009). Boose et al. (2016b) found a correlation of IN
activity with the quartz concentration in dust samples, which they collected either after being airborne and transported or
directly at the surface from deserts worldwide, suggesting that quartz particles have the potential to be relevant INPs for
cloud glaciation in the atmosphere. Indeed, quartz particles showed IN activity in laboratory studies, albeit with very
different IN efficiencies (Pruppacher and Sänger, 1955; Isono and Ikebe, 1960; Zimmermann et al., 2008; Atkinson et al.,
2013; Zolles et al., 2015; Kaufmann et al., 2016). Some early studies including Pruppacher and Sänger (1955) and Isono and
Ikebe (1960) found quartz to be IN active in their experiments. For supermicron quartz particles immersed in pure water,
Zimmermann et al. (2008) reported an activated fraction of 1 % at 261 K ($RH_w$ > 100%). In droplet freezing experiments,
Atkinson et al. (2013) and Zolles et al. (2015) found 50 % of droplets frozen as high as 249 K and as low as 235 K,
depending on sample origin and pretreatment (e.g. milling). These examples show that albeit the simple chemical
composition, the quartz surface seems to show large variations with respect to its surface properties resulting in highly
variable IN activities.

Mineral surfaces may undergo changes due to interaction with atmospheric chemical species while being transported over
long distances (Prospero, 1999; Schepanski et al., 2009; Uno et al., 2009). These changes can potentially alter their IN ability
(Salam et al., 2007; 2008; Kulkarni et al., 2012; Augustin-Bauditz et al., 2014). Several studies reported an indifferent
behavior of mineral surfaces to dissolved species so that immersion freezing in solutions can be simply described as a
freezing point depression due to the solute (Zuberi et al., 2002; Zobrist et al., 2008; Rigg et al., 2013), with a constant offset
in water activity ($\Delta a_w$ = const.), similar to the water-activity-based description of homogeneous IN by Koop et al. (2000).



This description was further elaborated as activity-based immersion freezing model (ABIFM) by Knopf and Alpert (2013).
In contrast, other studies have shown that IN efficiencies in immersion mode deteriorated due to irreversible surface destruction, e.g. in the presence of acids (Augustin-Bauditz et al., 2014; Wex et al., 2014; Burkert-Kohn et al., 2017).

This is the second part of three companion papers on IN activity of silicates and aluminosilicates. In Part 1 (Kumar et al. (2018a) we have shown that immersion freezing onset temperatures of microcline in aqueous solutions strongly deviate from a constant $\Delta a_w$. The observed deviations were both, to higher and lower IN temperatures, depending on solute type and
concentration. This finding is in accordance with Whale et al. (2018), who found an increase in IN activity for the K-feldspars microcline and sanidine in dilute $(NH_4)_2SO_4$ solutions, but a decrease in the presence of dilute NaCl.

In this paper and Part 3 (Kumar et al., 2018b) of the companion papers, we relate IN activities of mineral surfaces more closely with the mineral surface properties by investigating the differences in IN activity of structurally similar minerals in pure water and aqueous solutions. In Part 3 we investigate the differences in IN activities of aluminosilicates and whether the
enhancement observed for microcline in dilute $NH_3/NH_4^+$ containing solutions (shown in Part 1) is a more general property of aluminosilicates. In the current study, we present immersion freezing experiments of silicas (crystalline quartz and amorphous silicas) in pure water and in solution droplets in order to investigate their IN activity and how it is influenced by the presence of $NH_3$ and several inorganic salts, namely $(NH_4)_2SO_4$, $NH_4HSO_4$, $Na_2SO_4$ and NaOH. To elucidate which surface structures provide IN activity, we compare the IN activity of quartz with other silica particles, assuming that IN does
not occur on the whole particle surface with a uniform probability but that the surface exhibits active sites, i.e. preferred locations for IN with areas of $10 - 50$ nm$^2$ based on estimates using classical nucleation theory (Vali, 2014; Vali et al., 2015; Kaufmann et al., 2017).

Quartz is one of the most abundant minerals in the Earth's crust. Since quartz is of relevance to different scientific fields such as material sciences, geochemistry and chemical engineering, quartz surface properties and processes have been the
subject of many scientific studies. The dissolution and crystallization of quartz has gained great attention because it influences geochemical processes, such as the formation of mineralized deposits or the silica concentration in natural and industrial waters (Crundwell, 2017). Dry applications of quartz powders is of concern because of the pathogenicity of the ground particles (Fubini et al., 1989). We make use of the detailed characterization resulting from such studies to relate the IN activity of quartz to its surface structure.

**2 Methodology**

**2.1 Mineralogy, size distribution, milling and BET surface area measurements**

*Mineralogy.* Silica (three dimensional polymeric network of $SiO_2$) can exist in many different forms that can be crystalline as well as amorphous. Amorphous silica shows only short range order and lacks a crystalline structure as shown via X-ray diffraction (XRD) measurements (Poulsen et al., 1995). Quartz, most common form of crystalline silica, has a continuous
regular framework of tetrahedral $SiO_4$ units with Si in the center and oxygen atoms at the tetrahedral corners. Each oxygen atom is shared between two tetrahedra. It is a hard mineral (Mohs hardness 7) with no preferred cleavage plane (owing to the roughly equal bond strengths throughout the crystal structure) and typically breaks with conchoidal fracture. Quartz is the last mineral to crystallize from a magma i.e. it crystallizes at lower temperatures compared to other minerals, and therefore, it grows to fill the spaces remaining between the other crystals in form of a common impurity (Bowen, 1922, 1928).

*Size distribution.* Quartz from Sigma-Aldrich (~ 99 %) was the primary sample used in this study (see Sect. 4.4.5 for a discussion of the 1 % of impurities). We will refer to this sample in the following as SA quartz. As per manufacturer, SA quartz is a naturally occurring microcrystalline silica which has been finely ground resulting in a particle size range of 0.5 –



10 μm (approx. 80% between 1 – 5 μm). In addition, we determined the number size distribution with a TSI 3080 scanning mobility particle sizer (SMPS) and a TSI 3321 aerodynamic particle sizer (APS). Two lognormal distributions were fitted to the bimodal size distribution yielding mode diameters of 482 nm and 1.52 μm (see Supplementary Material). Scanning electron microscopy (SEM) measurements were also performed on the quartz sample at the ScopeM facility at ETH Zurich. The quartz samples were prepared by placing them on a graphite plate and coated with Pt/Pd alloy by sputtering before taking the images (see Supplementary Material). SEM measurements mostly concur to the size range provided by the manufacturer with the exception of the presence of very few larger particles (> 10 μm in diameter), which are not measured by APS since they are not well aerosolized owing to their higher mass. We also determined the mineralogical composition of the SA quartz sample by means of XRD in order to assess the mineralogical purity of the mineral. A quantitative analysis was performed with the AutoQuan program, a commercial product of GE Inspection Technologies applying a Rietveld refinement (Rietveld, 1967, 1969). Based on the X-ray diffractogram, the sample of SA quartz consists of 98.9 % (± 0.29 %) quartz, mixed with kaolinite (0.32 % ± 0.2 %) and topaz (0.76 % ± 0.2 %). The amorphous content is estimated as 4.5 ± 0.5 %. The Brunauer–Emmett–Teller (BET) nitrogen adsorption method was used to determine the specific surface area of this quartz sample as 4.91 $m^2$ $g^{-1}$. In addition, thermogravimetric analysis (TGA) was performed on the quartz sample to assess the presence of volatile species. Only a mere 0.30 % loss in weight was observed in TGA up to 350 °C (see Supplementary Material).

*Additional milling.* To assess the effect of further milling on the IN efficiency of the SA quartz sample, we milled a portion of the sample with a tungsten carbide disc mill for 40 s (with 1 min gap after the first 20 s of milling) before running emulsion freezing experiments.

*Other samples.* For comparison, we also used four other silica dusts besides SA quartz, namely:

(a) The quartz sample (BET value 3.67 $m^2$ $g^{-1}$) which showed little IN activity in Kaufmann et al. (2016) (termed Kaufmann quartz) (see Appendix A for details and explanation for its previously reported low IN activity). 0.17 % loss in weight was observed in TGA up to 350 °C (see Supplementary Material). The amorphous content is estimated as 6.4 ± 0.5 %.

(b) A crystalline quartz sample procured from the Technical University of Vienna (termed TU Vienna quartz), which is "quartz I" of Zolles et al. (2015). 0.2 % loss in weight was observed in TGA up to 350 °C (see Supplementary Material).

(c) Amorphous silica particles procured from Alfa Aesar (particle size: 0.4 – 0.6 μm (characterization by manufacturer), BET: 5.72 $m^2$ $g^{-1}$). In order to assess the effect of milling on the IN efficiency of amorphous silica, we milled a portion of this sample with a tungsten carbide disc mill for 40 s (with 1 min gap after the first 20 s of milling).

(d) Non-porous, amorphous silica particles procured from Zurich University of Applied Sciences (named Stöber particles in this article, mean particle diameter: 0.34 ± 0.02 μm; BET: 11 $m^2$ $g^{-1}$, see Supplementary Material for synthesis procedure).

Please note that both Alfa Aesar and Stöber silica particles are synthetically grown (using tetraethyl orthosilicate (TEOS) in alkaline conditions) samples used in this study and have not been milled. Only when indicated we milled the Alfa Aesar silica particles on purpose.

**2.2 Emulsion freezing of quartz freshly suspended in pure water or solutions**





Immersion freezing experiments were carried out with the differential scanning calorimeter (DSC) setup (Q10 from TA Instruments). 5 wt% SA quartz suspensions in water (molecular biology reagent water from Sigma Aldrich) were prepared in borosilicate glass vials with varying solute concentrations (0 – 20 wt%) viz. $(NH_4)_2SO_4$ (Sigma Aldrich, $\geq$ 99 %), $NH_4HSO_4$ (Sigma Aldrich, $\geq$ 99.5 %), $Na_2SO_4$ (Sigma Aldrich, $\geq$ 99 %), NaOH (Fluka Chemical, $\geq$ 99 %), $NH_3$ solution (Merck, 25

%). For comparison, 5 wt% TU Vienna quartz and a concentrated (8 – 9 wt%) Kaufmann quartz sample in water, as well as 10 wt% Stöber silica and Alfa Aesar silica suspensions in water containing $NH_3$ and $(NH_4)_2SO_4$ were also prepared for freezing experiments (See Appendix A for freezing experiment details for the Kaufmann quartz sample).

To avoid particle aggregation, the suspensions prepared in pure water or solutions were sonicated for 5 min before preparing the emulsions. The aqueous suspension and an oil/surfactant mixture (95 wt% mineral oil (Sigma Aldrich) and 5 wt% lanolin

(Fluka Chemical)) were mixed in a ratio of 1:4 and emulsified with a rotor-stator homogenizer (Polytron PT 1300D with a PT-DA 1307/2EC dispersing aggregate) for 40 seconds at 7000 rpm. This procedure leads to droplet size distributions peaking at about 2 – 3 μm in number and a broad distribution in volume with highest values between 4 and 12 μm similar to the size distributions shown in Figs. 1 of Marcolli et al. (2007), Pinti et al. (2012) and Kaufmann et al. (2016). We placed 4 – 8 mg of this emulsion in an aluminum pan, which was hermetically closed and subjected to three freezing cycles in the DSC

following the method developed and described by Marcolli et al. (2007). The first and the third freezing cycles were executed at a cooling rate of 10 K min$^{-1}$ to control the stability of the emulsion. The second freezing cycle was run at 1 K min$^{-1}$ cooling rate and used for evaluation (Zobrist et al., 2008; Pinti et al., 2012; Kaufmann et al., 2016; Kumar et al., 2018a).

The DSC registers the heat release when emulsion droplets freeze. When emulsions are prepared from an aqueous

suspension of INPs, the larger droplets are expected to freeze heterogeneously because they contain many particles while the smaller ones rather freeze homogeneously because they contain only few or no particles. Typical DSC thermograms therefore contain a freezing peak below about 237 K due to homogeneous IN, while freezing above this temperature is due to heterogeneous IN. The onset temperatures of the heterogeneous freezing peak ($T_{het}$) and the homogeneous freezing peak ($T_{hom}$) were determined as the intersection of the tangent drawn at the point of greatest slope at the leading edge of the peak

with the extrapolated baseline, whereas the melting temperature ($T_{melt}$) was determined as the maximum of the ice melting peak (see Fig. 1 of Kumar et al. (2018a)). The heat release is approximately proportional to the volume of water that froze heterogeneously or homogeneously and is represented by the integral of the peak over time. Note that this proportionality is only approximate because the enthalpy of freezing exhibits a temperature dependence (Speedy, 1987; Johari et al., 1994). We quantified the heterogeneously frozen fraction, $F_{het}$, as the ratio of the heterogeneous freezing signal to the total freezing

signal of the thermogram in the time domain (see Kumar et al. (2018a) for details). The evaluation of $F_{het}$ does not include the spikes that occur before the appearance of the heterogeneous freezing signal. These spikes originate from single droplets in the tail of the droplet size distribution (mostly between 100 – 300 μm with some up to 500 μm in diameter) being orders of magnitude larger in volume than the average droplets and not representative for the sample. Freezing experiments were performed on emulsions prepared from at least two separate quartz suspensions for each solute concentration and means are

reported. Average precisions in $T_{het}$ are $\pm$ 0.2 K with maximum deviations not exceeding 0.9 K (i.e. uncertainties slightly higher than in Kumar et al. (2018)). $T_{hom}$ and $T_{melt}$ are precise within $\pm$ 0.1 K. Absolute uncertainties in $F_{het}$, are on average $\pm$ 0.02 and do not exceed $\pm$ 0.12. However, $F_{het}$ carries larger uncertainties (up to $\pm$ 0.19) when heterogeneous freezing signals are weak or overlap (forming a shoulder) with the homogeneous freezing signal (as was the case for e.g. amorphous silica in various solutions and quartz aged in NaOH solution).

**2.3 Aging and reversibility of quartz suspended in water or solutions**



SA quartz (5 wt%) suspended in pure water, NH$_3$ solution (0.005 molal; pH 7.9; $a_w \approx 0.999$), (NH$_4$)$_2$SO$_4$ solution (10 wt%; pH 5.5; $a_w \approx 0.963$), NH$_4$HSO$_4$ solution (2 wt%; pH 1.1; $a_w \approx 0.988$), Na$_2$SO$_4$ solution (5 wt%; pH 6.8; $a_w \approx 0.986$) or NaOH solution (5x10$^{-3}$ molal; pH 9.5; $a_w \approx 1$ and 5x10$^{-6}$ molal; pH 7.1; $a_w \approx 1$) were aged in borosilicate glass vials and tested over a period of five days. Immersion freezing experiments were carried out with the DSC setup with emulsions prepared from at least 2 separate aging experiments for each solute concentration. Measurements were done on the day of preparation (fresh) and on the subsequent five days in order to assess the evolution and long-term effect of these solutes on the IN efficiency of quartz. Aging experiments conducted in pure water, NH$_3$ solution (0.005 molal) and NaOH solution (5x10$^{-3}$ molal) were repeated in polypropylene falcon tubes to assess the influence of leached contaminants from the borosilicate glass vials on the IN activity of quartz.

After aging for five days the suspensions were centrifuged for 2 minutes at 600 rpm, the supernatant solution was removed and the settled particles were washed with pure water. This process was repeated five times and the washed particles were resuspended in pure water and the IN efficiency of emulsions prepared from these suspensions were tested with the DSC setup.

In order to assess the leaching of contaminants from quartz and the vial walls, suspensions of quartz were prepared in pure water in both borosilicate glass vials and polypropylene falcon tubes and aged for 72 hours. The freshly prepared and aged suspensions were centrifuged to remove the particles. The supernatant liquid in each case was collected and tested for the concentration of leached elements via inductively coupled plasma mass spectroscopy (ICP-MS). The results of ICP-MS measurements are summarized in Supplementary Material (Sect. S10).

**3 Results**

**3.1 Ice nucleation activity of quartz and amorphous silica in pure water**

Figure 1 shows the DSC thermograms of suspensions of quartz and amorphous silica particles in pure water prepared as emulsion droplets. $F_{het}$ and $T_{het}$ are listed in Table 1. As can be easily seen, there are large differences in IN activities between the samples, from barely to highly IN active.

3.1.1 Quartz

In the emulsion freezing experiments, all investigated quartz samples clearly show IN activity. TU Vienna quartz is slightly more IN active (with respect to both, $T_{het}$ and $F_{het}$) than SA and Kaufmann quartz at similar suspension concentrations. $T_{het}$ of TU Vienna quartz (250.9 K) is slightly lower than the freezing onset reported by Zolles et al. (2015) (~ 252 K). We ascribe this difference to the higher quartz surface area present in the Zolles et al. (2015) droplet freezing setup compared to our emulsion freezing experiments. In accordance with Zolles et al. (2015), we observe an enhancement in IN efficiency of quartz due to milling (Table 1).

The quartz sample from Kaufmann et al. (2016) is clearly IN active, exhibiting a distinct heterogeneous freezing signal with $T_{het} \approx 242$ K and a shoulder extending to higher temperature with $T_{het} \approx 250$ K. Kaufmann et al. (2016) reported a heterogeneous freezing onset temperatures of ~ 247 K for the same sample, yet, with a very weak heterogeneous freezing signal corresponding to an IN active particle fraction of only 0.01. We explain in Appendix A that this was due to an underestimation of the coarse particle fraction because of the presence of very large particles (> 20 μm), which were not accounted for in the particle size distribution determined by SMPS/APS leading to a bias resulting in a too low estimate of the IN active fraction of quartz particles.

3.1.2 Amorphous silica particles



DSC thermograms of both amorphous silica samples (Fig. 1) show only a single freezing peak with onsets of 237.2 K and

237.1 K for Stöber and Alfa Aesar silica, respectively, listed under $T_{het}$ in Table 1. This freezing temperature is slightly higher but still within the uncertainty range of $T_{hom}$ of pure water emulsions (i.e. 237.0 K). Note that due to the volume dependence of homogeneous IN rates, $T_{hom}$ of the quartz samples is slightly lower than $T_{hom}$ of the pure water emulsions. Only the smallest droplets of the emulsified quartz suspensions are empty and therefore freeze homogeneously at a lower temperature than the larger droplets which give rise to the onset of the homogeneous freezing peak in pure water emulsions.

The calculation of the heterogeneously frozen fraction for both amorphous silica samples assumes that the heat signal at $T >$ 237.0 K is due to heterogeneous freezing, leading to $F_{het}$ below the uncertainty limits (see Table 1). Since both samples consist of submicron particles which should be well distributed between emulsion droplets (droplet size needed to incorporate on average one silica particle is ~ 1.1 μm in diameter), a prevalence of empty droplets cannot be the reason of the absence of detectable IN activity. Therefore, based on our emulsion freezing experiments we consider the amorphous silica

particles as inactive or barely IN inactive in water.

When the Alfa Aesar sample is milled, a clearly visible heterogeneous freezing signal develops consisting of two shoulders on the warmer end of the homogeneous freezing peak with onsets of 246.8 K and 239.7 K, again demonstrating the fundamental importance of the milling process.

### 3.2 Dependence of the heterogeneous freezing temperatures on the presence of solutes

For freezing experiments in the presence of solutes we concentrate on the SA quartz. The mean heterogeneous freezing onset ($T_{het}$), homogeneous freezing onset ($T_{hom}$) and ice melting temperatures ($T_{melt}$) for 5 wt% SA quartz suspensions in water and aqueous solution droplets are shown in Fig. 2a as a function of the solution water activity ($a_w$). The $a_w$ is obtained from the evaluation of the melting point depression measured during the heating cycle using the Koop et al. (2000) parameterization. Hence all melting temperatures lie exactly on the melting curve, except in case of Na$_2$SO$_4$ where above the eutectic

concentration of 4.6 wt% a hydrate of Na$_2$SO$_4$ crystallizes together with ice and $a_w$ had to be calculated based on the solute concentration using the AIOMFAC thermodynamic model at 298 K (Zuend et al., 2008; 2011). The measured $T_{hom}$ follows a similar $a_w$ dependency as $T_{melt}$ and has been parameterized by Koop et al. (2000). We construct this line by a constant shift of the melting curve by $\Delta a_w^{hom}(T) = 0.30$ (dotted black line) derived using the averaged $T_{hom}$ of all experiments of this study (which is in good agreement with $\Delta a_w^{hom}(T) = 0.305$ reported by Koop et al. (2000)). Similarly, we apply a constant offset

$\Delta a_w^{het} = 0.221$ to shift the ice melting curve to the heterogeneous freezing temperature of pure water, yielding the solid black line, which for simplicity will be referred to as $T_{het}^{\Delta a_w^{het}}(a_w)$ from here onwards (see Kumar et al. (2018a) for more details).

Analogous to the parameterization for $\Delta a_w^{hom}(T)$ based on the thermodynamic homogeneous IN description of Koop et al. (2000), $T_{het}^{\Delta a_w^{het}}(a_w)$ assumes that the water activity dependence of $T_{het}$ is determined by solute-driven changes in the structure of the water alone, while interactions of the solute with the INP surface are excluded. From Kumar et al. (2018a; 2018b) we

know that the assumption of $\Delta a_w^{het}$ fails when there are specific interactions between the solute and the mineral surface. As can be seen from Fig 2a the measured heterogeneous freezing onset temperatures, $T_{het}$, follow $T_{het}^{\Delta a_w^{het}}(a_w)$ within measurement uncertainties for quartz suspensions in (NH$_4$)$_2$SO$_4$, NH$_4$HSO$_4$ and Na$_2$SO$_4$, but fall below this line in the presence of the bases NH$_3$ and NaOH. In these alkaline solutions, $T_{het}$ for quartz emulsions strongly falls below $T_{het}^{\Delta a_w^{het}}(a_w)$ at dilute solute concentrations ($a_w \geq 0.99$) and stay almost parallel to $T_{het}^{\Delta a_w^{het}}(a_w)$ to higher solute concentrations.

This decrease in $T_{het}$ is less pronounced in the presence of NH$_3$ than for suspensions in NaOH.

### 3.3 Dependence of the heterogeneously frozen fraction on the presence of solutes



While the addition of neutral or acidic solutes does not influence the freezing temperature beyond the expected freezing point depression described by $T_{het}^{\Delta a_w^{het}}(a_w)$, it does affect the heterogeneously frozen fraction. Figure 3 shows the DSC thermograms for emulsion freezing of 5 wt% SA quartz suspended in increasingly concentrated $(NH_4)_2SO_4$ solutions. The

dotted brown line connecting the onsets of the heterogeneous freezing signals depicts the continuous decrease in $T_{het}$ as the $(NH_4)_2SO_4$ concentration increases. An increase in heterogeneous-to-homogeneous freezing ratio with increasing solute concentration is apparent. $F_{het}$ increases up to $a_w = 0.998$ (0.5 wt%) and stays around this increased value when the solute concentration is further increased. Figure 2b shows the evaluation of the freezing signals in terms of the heterogeneously frozen fraction $F_{het}$, as a function of $a_w$ for all investigated solutes. For solutions containing the salts $(NH_4)_2SO_4$, $NH_4HSO_4$,

and $Na_2SO_4$, $F_{het}$ shows a constant increase compared with the pure water case, albeit hardly exceeding the maximum uncertainty limit. Despite the decrease in $T_{het}$, there seems to be a slight increase in $F_{het}$ for quartz suspended in aqueous solutions with higher $NH_3$ ($a_w \geq 0.98$) concentrations. On the other hand, even very low concentrations of NaOH ($a_w \leq 0.99$) strongly decrease $F_{het}$. The heterogeneously frozen fraction slightly recovers at higher concentrations of NaOH ($a_w = 0.99 – 0.96$) yet remains significantly below the pure water case.

Furthermore, in Part 3 (Kumar et al., 2018b) we show that micas (muscovite and biotite) and gibbsite, which reveal no IN activity in pure water emulsion freezing experiments, develop a heterogeneous freezing signal in the presence of $NH_3$ and $(NH_4)_2SO_4$. We therefore tested the amorphous silica samples for a similar effect. DSC thermograms of both amorphous silica samples suspended in $NH_3$ (0.05 molal and 0.5 molal) and $(NH_4)_2SO_4$ (0.05 wt% and 1 wt%) solutions, corresponding to an $a_w$ range of $1 – 0.987$ (Fig. 4), show only one clear freezing signal. We report the onset of this freezing peak as $T_{het}$ in

Table 2, while under $T_{hom}$ we list the onset freezing temperature of the reference solutions (prepared with the solute only). Neither the Alfa Aesar nor the Stöber silica particles (10 wt% suspensions) show a significant increase in the freezing onset compared with the reference measurements. We evaluate the heterogeneously frozen fraction by attributing the freezing signal at temperatures above the reported $T_{hom}$ of the reference measurement to heterogeneous freezing, yielding $F_{het}$ within the uncertainty range (see Table 2). We therefore conclude that $NH_3$ and $(NH_4)_2SO_4$ do not lead to a discernable

enhancement of the IN activity of amorphous silica.

### 3.4 Aging and recovery experiments of quartz in water and aqueous solutions

In order to assess the long-term effect of solutes on the IN efficiency of quartz, aging experiments were performed over a period of 5 days with 5 wt% SA quartz suspensions in pure water (prepared in borosilicate glass vials and polypropylene falcon tubes) and various inorganic solutes. Every day, aliquots were taken from the suspension and tested in emulsion

freezing experiments. For these experiments, $T_{het}$ and $F_{het}$ are given in the upper and lower panels of Figs. 5 and 6, respectively. Figure 5 shows the results of experiments performed in glass vials while Fig. 6 compares $T_{het}$ and $F_{het}$ of experiments performed in glass vials and polyprolylene tubes. After aging, the quartz suspensions were decanted, washed and resuspended in pure water in order to assess the reversibility of any surface modification occurring during the aging period. Figures 5 and 6 also show the change in $T_{het}$ and $F_{het}$ when aged quartz is resuspended in pure water. (Note that $T_{het}$

and $F_{het}$ in the experiments with fresh dusts can be lower than in the reversibility tests, because the former were performed in solutions and the latter in pure water.)

Aging of quartz suspensions in pure water in glass vials decreases the IN activity in terms of both, $T_{het}$ and $F_{het}$. Strong decreases in IN efficiency occur during aging in the presence of $Na_2SO_4$, NaOH and $NH_3$. On the other hand, $T_{het}$ and $F_{het}$ remain constant when quartz is suspended in 2 wt% $NH_4HSO_4$ solution. Repetition of the aging experiments reveals

considerable variability in the decrease of IN efficiency over time, as indicated by the large min-to-max bars after $2 – 5$ days. The reversibility tests show complete or almost complete recovery of IN efficiency after washing in the case of aging in pure





water and in solutions containing $Na_2SO_4$, NaOH (except at high pH 9.5) and $(NH_4)_2SO_4$. Interestingly, instead of recovering, the IN efficiency after aging in a dilute $NH_3$ solution decreases even further when the particles were washed and resuspended in pure water.

To elucidate whether leached material from the surface of the glass vial had any impact on the decrease of IN activity during aging of quartz in glass vials, experiments exhibiting a strong decrease in IN efficiency were repeated in polypropylene falcon tubes. Indeed, in contrast to aging in glass vials, quartz aged in pure water in polypropylene tubes exhibited a stable $T_{het}$ and a slight enhancement in $F_{het}$ (yet within the uncertainty range) (Fig. 6). When quartz is suspended in a dilute $NH_3$ (pH 7.9) solution in polypropylene tubes, similar trends were observed as in glass vials, with constant $T_{het}$ after an initial

decrease, followed by a further decrease after washing with pure water. Interestingly, quartz freshly suspended in NaOH (pH 9.5) in polypropylene tubes shows a stronger initial decrease in IN efficiency ($T_{het}$ and $F_{het}$) but stays higher from the second day onwards than in the corresponding experiments performed in a glass vial until the end of the experiment.

Another set of SA quartz suspensions (5 wt%) was prepared in pure water in glass vials and aged for 7 months to investigate the aging effect over very long time scales. Figure 7 shows that the aged particles almost completely lost their IN efficiency

and barely any of it was recovered after the aged particles were washed and resuspended in pure water. In Section 4.4, we relate the results of the aging and reversibility experiments to surface processes occurring in the different solutions.

## 4 Discussion

### 4.1 IN efficiency of quartz and amorphous silica particles in pure water

As shown in Fig. 1 and Table 1, the IN activity of quartz is superior to the one of amorphous silica particles. In contrast to

results reported in Kaufmann et al. (2016), all quartz samples investigated in this study including Kaufmann quartz proved to be highly IN active. Reanalysis of the quartz sample used in Kaufmann et al. (2016) revealed a large contribution of coarse particles to the sample mass resulting in a prevalence of empty emulsion droplets even at high suspension concentrations, contributing to the homogeneous freezing signal (see Appendix A for a detailed discussion).

The freezing onset temperatures in pure water for the investigated quartz samples range from 247 K to 251 K (see Table 1),

covering a similar temperature range as the quartz samples investigated by Atkinson et al. (2013) and Zolles et al. (2015). On the other hand, the IN activity of the silica particles (Stöber and Alfa Aesar) is negligible. Hardly any IN activity of amorphous silica particles is in agreement with Zobrist et al. (2008) who observed freezing at around 255 K in bulk freezing experiments with 3 μl droplets containing $10^9 – 10^{10}$ particles (total mean particle surface area $0.25 – 2.5$ $cm^2$). This freezing temperature is close to the one of pure water droplets of this size ($252 – 253$ K). Whale et al. (2018) found freezing onsets of

silica particles from Sigma-Aldrich (silica gel) above 262 K, however, for freezing experiments with larger surface areas of silica present in a droplet (~ 10 $cm^2$).

Zolles et al. (2015) investigated the density of IN active sites of three quartz samples and found a very high variability from hardly IN active to an activity similar to that of microcline. Their quartz sample from Sigma-Aldrich corresponding to the TU Vienna quartz sample in this study was the most IN active and a natural quartz sample the least IN active. Moreover, the

IN activity of the natural quartz sample increased considerably upon milling. Our comparison of emulsion freezing experiments with SA quartz additionally milled and original SA quartz (used as obtained from the manufacturer) corroborate an increased IN efficiency in terms of $T_{het}$ and $F_{het}$ for the additionally milled sample as shown in Fig. 1 and Table 1. Moreover, milling of the amorphous silica sample from Alfa Aesar also had a very positive effect on its IN activity.

### 4.2 Heterogeneous IN of quartz in aqueous solutions





The water-activity-based description predicts heterogeneous IN temperatures ($T_{het}^{\Delta a_w^{het}}(a_w)$) as a function of $a_w$ by shifting the ice melting curve by a constant offset in $a_w$. It is expected to be valid in the absence of specific interactions between the solute and the ice-nucleating surface so that the only effect of the solute is a freezing point depression. Such a description has been suggested by several studies in the recent past (Zuberi et al., 2002; Archuleta et al., 2005; Cantrell and Robinson, 2006; Zobrist et al., 2006; Zobrist et al., 2008; Alpert et al., 2011b; Alpert et al., 2011a; Knopf and Forrester, 2011; Rigg et al.,
350    2013).

In Part I of this series, we showed that heterogeneous freezing onsets of microcline showed strong deviations from the water-activity-based description to higher freezing temperatures from predicted values when suspended in dilute $NH_3/NH_4^+$ containing solutions and substantial decrease in IN efficiency in more concentrated solutions of inorganic salts including $NH_4^+$ containing salts and $NH_3$ (Kumar et al., 2018a). In Kumar et al. (2018b) we extended this investigation to other
aluminosilicates and found that an increase of $T_{het}$ in the presence of $NH_4^+$ containing solutes is a general feature of feldspars, clay minerals and micas, while the decrease in IN efficiency at higher solute concentration is a more specific characteristic of K-feldspars and most pronounced for microcline. Conversely, SA quartz follows quite well the water-activity-based prediction of $T_{het}$ in case of pH neutral suspensions. $F_{het}$ shows an initial increase in very dilute solutions of $(NH_4)_2SO_4$, $NH_4HSO_4$ and $Na_2SO_4$, which is preserved to higher concentrations (Fig. 2b). Whale et al. (2018) compared the IN activity
of quartz in dilute NaCl and $(NH_4)_2SO_4$ with the one in pure water and observed no change of the freezing onset temperatures in the presence of the solutes, but an increase in the density of IN active sites towards lower temperatures in a dilute $(NH_4)_2SO_4$ solution and a decrease in a dilute NaCl solution.

Opposite to the effect of $NH_3$ on aluminosilicates, there is a decrease of $T_{het}$ for SA quartz in $NH_3$ solutions. In the past, several infrared spectroscopy studies have investigated the adsorption of $NH_3$ molecules on various types of mineral oxides
(Mapes and Eischens, 1954; Eischens and Pliskin, 1958; Peri and Hannan, 1960). The quartz surface provides several centers for interaction with $NH_3$ molecules viz. (i) hydrogen-bonding via one of its hydrogen atoms with a surface oxygen atom of a silanol groups; (ii) hydrogen-bonding via its nitrogen atom with the hydrogen of a surface hydroxyl group; (iii) coordination to an electron-deficient Si (Lewis acid site) (Folman, 1961; Cant and Little, 1965; Blomfield and Little, 1973; Tsyganenko et al., 1975; Morrow and Cody, 1976; Morrow et al., 1976; Fubini et al., 1992; Li and Nelson, 1996; Wright and Walsh, 2012).
In addition to reversible coordination to silanols via hydrogen bonding, $NH_3$ may interact irreversibly with strained siloxane bridges by disrupting them into $Si-NH_2$ and $Si-OH$ groups (Folman, 1961; Peri, 1966), although, water displays more affinity than $NH_3$ for this reaction (Blomfield and Little, 1973; Morrow and Cody, 1976; Fubini et al., 1992). Figure 2 shows that the sum of these interactions seems to affect the IN activity of the quartz surface by decreasing $T_{het}$ but slightly increasing $F_{het}$ at higher $NH_3$ concentrations. Using sum frequency generation spectroscopy, Wei et al. (2002) have found an
enhancement of the hydrogen bonded OH peak in the presence of ammonia. They proposed that $NH_3$ molecules may form strong hydrogen bonds with the silanol groups on the silica surface with the nitrogen atoms facing silica, resulting in an excessive number of protons (NH bonds) pointing into ice. However these interactions of $NH_3$ with the quartz surface do not seem to lead to an enhanced IN activity compared with the pure water case.

 Wright and Walsh (2012) found no strong and stable hydrogen bonding of $NH_4^+$ to hydroxylated quartz in their first
principles molecular dynamics simulations. The slight decrease in $T_{het}$ in the presence of $(NH_4)_2SO_4$ together with the slight increase in $F_{het}$ indicate that the presence of ammonium in the solution influences the IN activity, although only slightly.

The decrease of IN activity in NaOH containing suspensions as shown in Fig. 2 can be ascribed to the detrimental effect of alkaline conditions (pH between 7.1 and 13.6) on the stability of the quartz surface (House and Orr, 1992; Crundwell, 2017) and is discussed in the next section.



### 4.3 Aging effect and reversibility of surface modifications

The aging experiments performed with quartz suspended in different solutions point to surface processes that influence the IN activity of quartz over time. The IN activity was maintained over the whole aging period (5 days) in case of $NH_4HSO_4$ (2 wt%, pH 1.1). In contrast, in pure water, $(NH_4)_2SO_4$ (10 wt%, pH 5.5), $Na_2SO_4$ (5 wt%, pH 6.8) and very dilute NaOH ($5\times10^{-6}$ molal; pH 7.1), the IN activity decreased over time but was completely or almost completely restored after washing in water (Fig. 5). In more alkaline solutions, namely $NH_3$ (0.005 molal, pH 7.9) and more concentrated NaOH ($5\times10^{-3}$ molal, pH 9.5), IN activity was permanently lost (Fig. 6). Also, when quartz was aged for 7 months in pure water in glass vials, the IN activity was almost completely destroyed (Fig. 7). In contrast, the IN efficiency in pure water was barely affected during aging in polypropylene tubes over 5 days. A decrease of IN activity due to aging for 72 hours in water was also observed for Kaufmann quartz (Fig. A2) and TU Vienna quartz (Fig. A3). In the following, we relate the results of the aging and reversibility experiments to surface processes occurring in the different solutions.

### 4.3.1 Dissolution and growth of quartz in pure water

Several studies have discussed the effect of solution pH on dissolution rates of quartz (Henderson et al., 1970; Kline and Fogler, 1981; Schwartzentruber et al., 1987; Bennett et al., 1988; Knauss and Wolery, 1988). The dissolution rate of quartz at 25° C is $10^{-13} - 10^{-12}$ moles Si $m^{-2}$ $s^{-1}$ at low to neutral pH (0.5 – 7) and increases roughly linearly with increasing pH, reaching a value of $10^{-10}$ moles Si $m^{-2}$ $s^{-1}$ at pH 12 (House and Orr, 1992; Crundwell, 2017). The dissolution of quartz is considered to occur on deprotonated silanols, i.e. Si–O⁻ (Brady and Walther, 1989, 1990). Deprotonation of the silanol weakens the remaining siloxane bridges, facilitating the attack by water molecules and ultimately releasing the Si in the form of silicic acid ($H_4SiO_4$). Moreover, dissolution of quartz increases with increasing ionic strength of the solution, i.e. increasing salt concentration (Brady and Walther, 1990). At low pH, different dissolution mechanisms may be involved, such as $H_2O$ hydrolysis of Si centers or adsorption of $H^+$ onto siloxane bridging oxygen (Xiao and Lasaga, 1994; Criscenti et al., 2006; Bickmore et al., 2008), however with a low efficiency.

The SA quartz as provided by the manufacturer contains a minor share of amorphous material ($4.5 \pm 0.5$ %), produced most likely by grinding (Fubini et al., 1989). Because of the higher dissolution rate of amorphous silica compared to quartz, silicic acid should be released at a higher rate from the amorphous material (Crundwell, 2017). Since also the solubility of amorphous silica is higher (~50 ppm Si at 25 ℃) than the one of quartz (1 – 3 ppm Si at 25 ℃) (Walther and Helgeson, 1977), quartz should grow at the expense of amorphous silica over time when kept together in a closed vessel. After aging for 72 h in water (in a glass vessel), the amorphous fraction of the SA quartz sample indeed showed a slight decrease, however still within experimental error ($4.0 \pm 0.5$ % for the aged sample compared with $4.2 \pm 0.5$ % for the sample shortly (15 min) exposed to water). This is in agreement with the fact that the conversion of amorphous silica to quartz is a slow process.

ICP-MS measurements (see Tables S1 and S2; Supplementary Material) performed with the supernatant of a 0.5 wt% SA quartz suspension which was aged in pure water for 72 hours in a borosilicate glass vial shows a concentration of 11.9 ppm Si, which is well above the solubility of quartz in pure water and well below that of amorphous silica. The Si concentration after aging a 0.7 wt% SA quartz suspension for 72 h in polypropylene falcon tubes reaches only 4.9 ppm, indicating that a considerable fraction of the dissolved Si stems from the glass vial in the aged sample. Indeed, the glass vial continuously leaches Si to the water (Bunker, 1994). The ICP-MS measurement of pure water, which was in contact with the glass vial for less than an hour, was shows a concentration of only 0.2 ppm Si but reached 4.5 ppm Si after 72 h. This slower increase Si concentration compared to the immediate increase in case of the quartz samples (1.3 – 6 ppm after less than an hour in water) is simply due to the much smaller surface area of the glass vial (~2 orders of magnitude) exposed to water compared





with the one of the SA quartz sample. In addition, also the release of Si from the quartz sample is expected to stem mostly from its amorphous shares. Subsequently, silicic acid in water may form dimers, trimers and cyclic species due to autopolycondensation that sets in when the silicic acid concentration approaches the solubility limit (Perry, 1989, 2003; Belton et al., 2012). These oligomerization reactions are reversible (Tamahrajah and Brehm, 2016). We assume that at high Si concentration, silicic acid and its oligomers adsorb on the quartz surface, covering large parts of the crystalline surface or

at least a relevant fraction of the IN active sites, thus hampering the IN activity of the quartz samples aged in glass vials. Indeed, monolayer adsorption of silicic acid on quartz has been observed under conditions supersaturated with respect to crystalline quartz (Berger et al., 1994). Since leaching of Si from the glass vials is slow owing to the comparatively small exposed surface area, the freshly prepared suspensions of SA quartz in glass vials are not affected by an adsorbed layer, which is in accordance with our experiments (see Figs. 5 and 6). The Si concentration in the polypropylene tubes remains too

low to give rise to an adsorbed layer on the quartz surface even during aging and the IN activity is not hampered (see Fig. 6).

With time, the Si supersaturation with respect to quartz should lead to crystalline quartz growth. We assume that covalent bonds form between the adsorbed siliceous layer and the quartz surface leading to a grown, intact quartz surface with little defects. Since the emulsion freezing experiments of the SA quartz sample aged for 7 months in a glass vial show hardly any IN activity even after washing with pure water (see Fig. 7), we conclude that slowly grown quartz surfaces are indeed not

amorphous, but have a regular crystalline structure. These are barely IN active and only milling provides the quartz surface with IN active sites.

### 4.3.2 Dissolution and growth of quartz in solutions

Similar to the experiments in pure water, quartz suspended in glass vials at near-neutral conditions (i.e. $(NH_4)_2SO_4$ 10 wt%, $Na_2SO_4$ 5 wt% and NaOH $5 \times 10^{-6}$ molal) shows a decrease in IN efficiency, which is restored when the sample is washed and

resuspended in pure water. This suggests the formation and washing off of a siliceous layer similar as in the pure water case.

Under alkaline conditions both, quartz dissolution and growth rates are increased. Indeed, centimeter-sized synthetic quartz crystals are grown from amorphous silica on seed crystals at high temperatures and pressures in highly alkaline conditions (Baughman, 1991). Such rigorous conditions are applied to accelerate the crystal growth. In the $5 \times 10^{-3}$ molal NaOH solution (pH 9.5), the solubility of amorphous silica and quartz are higher than at near-neutral conditions and dissolution and

growth of quartz are enhanced. We therefore ascribe the strong and immediate loss of IN activity of SA quartz suspended in $5 \times 10^{-3}$ molal NaOH solution in polypropylene tubes to the dissolution of quartz, which levels off when the equilibrium condition is approached. Therefore, during the following days of aging, there is no further decrease in IN activity, rather, $T_{het}$ and $F_{het}$ show a slight increase (however, within the uncertainty limits). After washing with water, $F_{het}$ recovers to the initial value in $5 \times 10^{-3}$ molal NaOH but remains clearly below the value measured in pure water. When the experiment is carried

out in glass vials (in $5 \times 10^{-3}$ molal NaOH), the initial loss of IN activity is less pronounced, probably because the additional leaching of Si from the glass vial leads to a quick increase of the Si concentration above the saturation level with respect to quartz. As a consequence, the quartz sample is in growth conditions for most of the aging time and we ascribe the irreversible loss of IN activity to growth of intact quartz layers which is faster than at neutral conditions because of the higher Si solubility under alkaline conditions.

Interestingly, during aging in $NH_3$, the main decrease of $T_{het}$ is observed after one day while $F_{het}$ is preserved during the five days of aging but is reduced when $NH_3$ is removed from the suspension. This implies that the presence of $NH_3$ can temporarily stabilize the surface followed by a strong decrease when it is removed.

### 4.4 Which factors determine the IN activity of quartz?




### 4.4.1 Crystallinity and substrate-ice lattice match

Crystallinity and lattice match between substrate and ice are often considered to provide IN activity to the substrate. While ice and silica exhibit structural similarities in form of tetrahedral building units, the most common hexagonal (Ih) and cubic (Ic) ice phases show structural analogies to the crystalline silica tridymite and cristobalite, respectively, but not to quartz (Tribello et al., 2010). Indeed, when the quartz surface is not produced by milling but by crystal growth, quartz is similarly inactive as amorphous silica. Conversely, when hardly active amorphous silica particles are milled, they become IN active.

This suggests that the regular quartz surface is not able to template ice growth and that the crystallinity of quartz is not a prerequisite for its IN activity.

### 4.4.2 Milling and radical site formation

Micrometer and nanometer sized quartz particles are usually obtained by milling. Due to the covalent nature of the quartz crystal lattice, considerable force needs to be exerted to obtain small particles by milling. The shear and compression applied

to quartz leads to a disturbed amorphous zone of $10 - 30$ nm thickness with dangling $Si–O^\bullet$ and $Si^\bullet$ radical sites that can be detected by electron paramagnetic resonance (EPR) (Fubini et al., 1987; Fubini et al., 1989; Makoto and Motoji, 1996). When quartz is ground in the atmosphere, the resulting $Si–O^\bullet$ and $Si^\bullet$ radical sites react with atmospheric species (Fubini et al., 1987). Reactions with $O_2$ lead to the formation of $Si–O–O^\bullet$, $Si–O–O–Si$ or $Si–O–O–O–Si$, which react in the presence of water (vapor) to $Si–OH$ and $Si–O–OH$. In addition, hydroxylation of strained $Si–O–Si$ groups (Fubini et al., 1987) results in

silanol groups ($\equiv Si–OH$) rendering the surface highly hydroxylated. Depending on their arrangement on the silica surface, silanols are isolated ($(Si–O)_3Si–OH$), germinal ($=Si(OH)_2$) or vicinal ($=Si(OH)–O–Si(OH)=$). The relatively non-polar siloxane bridges ($\equiv Si–O–Si\equiv$) may be strained and easily hydrolyzed to silanols or regular and hardly reactive (Morrow and Cody, 1976; Morrow et al., 1976; Brinker et al., 1986; Zhdanov et al., 1987; Brinker et al., 1988; Bolis et al., 1991; Fubini et al., 1992).

The IN activity observed for milled Alfa Aesar silica suggests that shear and compression applied to amorphous silica leads to similar radical sites as in the case of quartz, providing amorphous silica with IN activity. XRD analysis of Kaufmann quartz and SA quartz support the presence of amorphous shares of about 6.4 and 4.5 wt% in the samples, respectively. A part of this amorphous material may agglomerate as separate particles or in specific regions of the quartz surface. In the supplementary material, we show a collection of representative SEM images of SA and Kaufmann quartz that show

agglomerates on top of the quartz surface, which might be amorphous. Nevertheless, an amorphous surface layer on top of the quartz particles cannot be excluded. Because of the strong correlation of milling and IN activity of silica surfaces, we propose that the surface functionalization of silica particles arising from breaking covalent $Si–O$ bonds during milling gives rise to the IN activity of these materials.

Interestingly, the conjecture that surface functionalization resulting from the cleavage of covalent $Si–O$ bonds rather than the

ordered crystalline structure determines the surface properties of silica is confirmed by findings from a completely different research field. Ground quartz particles may induce silicosis, lung cancer and autoimmune diseases (Donaldson and Borm, 1998; Fubini, 1998). This pathogenicity is totally absent in chemically prepared amorphous silica or synthesized (grown) quartz particles. Since exposure to ground amorphous (vitreous) silica also leads to adverse health effects (Turci et al., 2016), the pathogenicity is considered to arise rather from the mechanical cleavage of the covalent $Si–O$ bonds than from the

crystallinity (Fubini et al., 1987; Fubini et al., 1989; Turci et al., 2016).

### 4.4.3 Surface OH groups





While siloxanes and silanols dominate the surfaces of amorphous silica and quartz, the relative surface densities of these groups show large variation. Fully hydroxylated quartz surfaces may carry up to 9.5 OH nm$^{-2}$ on the (001) surface and still 5.8 OH nm$^{-2}$ on the (011) and (101) surfaces (Musso et al., 2009). On the other hand, fully hydroxylated amorphous silica surfaces carry typically only 4.6 – 4.9 OH nm$^{-2}$. Highly hydroxylated surfaces are dominated by vicinal and geminal silanols, with few isolated silanol and siloxane groups (Zhuravlev, 2000; Muster et al., 2001). Due to their more ordered structure, silanols on quartz surfaces tend to form networks of chains of hydrogen bonds, whereas amorphous silica surfaces rather exhibit patches of hydrogen bonded silanols, even if their average silanol densities are the same (Musso et al., 2011).

When quartz and silica samples are heated, surface hydroxylation decreases due to the replacement of silanols by siloxanes. The more severe the heating conditions (temperature, vacuum, duration), the more dehydroxylized the surface becomes. Heating (calcination) to 670 K in vacuum removes all vicinal silanols of amorphous silica while isolated silanols are still present but become continuously scarcer by further heating (Zhuravlev, 2000). Quartz, on the other hand, is less easily dehydroxylated (Bolis et al., 1985). Dehydroxylated surfaces slowly rehydroxylate when they are exposed to humidity or in contact with liquid water. The Stöber particles shown in Fig. 1 have been heated to 823 K, which strongly decreased the number of vicinal silanols and subsequently hydrolyzed so that vicinal silanols should be restored to a full hydroxylation level of amorphous silica (i.e. 4.6 – 4.9 OH nm$^{-2}$).

Density functional theory based molecular dynamics (DFTMD) simulations by Sulpizi et al. (2012) showed that two main types of silanol groups occur at the surface of quartz, i) out-of-plane silanol groups forming short and strong hydrogen bonds with water molecules associated with a pK$_a$ value of 5.6; ii) in-plane silanol groups forming weak hydrogen bonds with the interfacial water molecules with a pK$_a$ value of 8.5. The former are referred to as "ice-like" water due to similarities in structure to water molecules in bulk ice and the latter as "liquid-like" water (Richmond, 2002). Musso et al. (2012) showed in an ab initio molecular dynamics study that the silanol surface of quartz (100) induces an ice-like structure of water in the proximity of the surface, which is more pronounced when the silanol density at the surface is higher. While water molecules spontaneously form hydrogen bonds to isolated silanols, hydrogen bonding to vicinal silanols involves breaking the surface hydrogen bond network between them. This is an activated process with an activation energy that increases with increasing length of the interconnected silanol chains (Musso et al., 2011). Water was only able to disrupt the weak internal hydrogen bonds between surface silanols with H···O > 2 Å but not the stronger ones with H···O < 2 Å (Musso et al., 2012). Milling decreases the long-range order of silanols and generates hydrophilic and hydrophobic patches (Turci et al., 2016). The generated defects may disrupt the chain of interconnected silanols and free them to participate in hydrogen bonding with water molecules. This could explain the enhanced IN activity of freshly milled quartz as well as freshly milled amorphous silica.

Recently, it had been suggested that the OH density and the substrate-water interaction strength are useful descriptors of a material's IN ability (Pedevilla et al. 2017). The example of quartz shows that OH density alone is indeed insufficient as a predictor for IN ability when strong hydrogen bonding amongst surface OH groups prevail over substrate-water interactions.

### 4.5.4 Protonation/deprotonation of surface OH groups and surface charge

Depending on the solution pH, the silanol groups protonate or deprotonate, thus changing the surface charge. The quartz surface is at the point of zero charge (PZC) around pH 2 and becomes more negative with increasing pH (Vidyadhar and Hanumantha Rao, 2007; Turci et al., 2016). At PZC, Si–OH groups prevail, and the number of Si–OH$_2^+$ equals the number of Si–O$^-$ groups. The ordering of water at the quartz surface was shown to be pH dependent. At pH 1.5 when the surface is slightly positively charged and at pH 12.3 when the surface is negatively charged, water molecules are ordered but the orientation is reversed from low to high pH. At intermediate pH, there is more disorder (Du et al., 1994; Richmond, 2002).





Zeta potential measurements show that the milled quartz surface is slightly less negatively charged than grown quartz surfaces at neutral conditions. However, milling increases the heterogeneity of the silanols and creates more acidic sites as indicated by a shallower increase of surface charge with decreasing pH (Turci et al., 2016).

Besides surface functionalization, surface charge is a factor that has been shown to influence IN activity (Marcolli et al., 2016; Abdelmonem et al., 2017; Kumar et al., 2018b). Abdelmonem et al. (2017) found that the freezing temperature of water at the $\alpha$-$Al_2O_3$ (0001) surface is highest when the surface is close to the PZC and the water molecules at the surface are least ordered. In contrast, Kumar et al. (2018b) shows that hydroxylated surfaces which had PZCs at low pH such as feldspars showed higher IN activity in pure water than surfaces with PZC shifted to neutral or alkaline conditions. These
contradictory results indicate that surface charge is not a reliable predictor of IN activity.

4.4.5 Impurities on mineral surfaces

Often IN activity of mineral surfaces has been related to the presence of impurities introducing special sites on the mineral surface (O'Sullivan et al., 2014; Zolles et al., 2015). Milling can accumulate impurities from within the crystal lattice (DeMott et al., 2003; Boose et al., 2016a) on the surface because regions rich in impurities provide preferred cleavage planes
(Whale et al., 2017). If such impurities are surface active and remain adsorbed on quartz when the particles are immersed in water, they can either block active sites and decrease the IN activity or generate new active sites. Milling of quartz leads to high energy surface sites that may attract semivolatile impurities to reduce the surface energy. Indeed, organic semivolatile material has been considered to provide IN activity to mineral surfaces (O'Sullivan et al., 2014; Tobo et al., 2014).

We therefore determined the presence of semivolatile material on the Kaufmann and SA quartz samples by performing TGA.
Since only a mere 0.17 % and 0.30 % loss in weight was observed in TGA up to 350 °C (Supplementary Material) in Kaufmann and SA quartz samples, respectively, the influence of semivolatile material on the IN efficiency is unlikely. To investigate the possibility that nonvolatile but water-soluble impurities provide IN activity to the quartz surface, we performed ICP-MS tests on the supernatant liquid of quartz suspensions (0.7 wt%) freshly prepared in water in polypropylene falcon tubes. Si had a contribution of more than 80 % of the leached elements in both quartz cases (see
Supplementary Material; Table S2 for details). The most abundant impurities were Na, Al, K, Ca, Ba, Fe and Co with a combined contribution of only 12 % and 18 % to the total leached elements for SA and Kaufmann quartz, respectively. Depletion of these elements from the quartz surface cannot explain the loss of IN activity during aging because the IN activity can be restored by washing the quartz particles with pure water. We therefore regard the abundance of Si-OH groups and their arrangement on the quartz surface relevant for IN activity in emulsion freezing experiments rather than the
presence of foreign components. However, we do not exclude that IN activity observed at higher temperatures in bulk freezing experiments might be due to impurities (O'Sullivan et al., 2014). Indeed, the notion that different types of sites are relevant at higher than at lower freezing temperatures is corroborated by Kaufmann et al. (2016) who showed in their Fig. 6 that only a weak correlation exists between freezing temperatures of bulk and emulsion freezing experiments of different mineral dusts.

From the above discussion, we conclude that milling is a requirement for quartz to be IN active and that IN occurs on specific active sites introduced by milling which are more reactive than the grown quartz surface. Despite the high degree of hydroxylation, the regular quartz surface does not give rise to IN activity, likely because most silanols are tightly interconnected by hydrogen bonds in a network that is too strong to be disrupted by water molecules.

**5 Atmospheric implications**





Arid and semiarid regions are the main sources of mineral dust (e.g. Saharan and Gobi Deserts) (Laurent et al., 2006; Laurent et al., 2008) which can have atmospheric lifetimes of several days (Huneeus et al., 2011). The dust, while being transported, can interact with a variety of trace gases (Usher et al., 2003; Kolb et al., 2010), which can lead to changes in the surface physicochemical properties. Not only ground-collected/near-source but also transported dusts have been reported to be rich in quartz (Avila et al., 1997; Alastuey et al., 2005; Field et al., 2006; Boose et al., 2016b; Kaufmann et al., 2016).

Quartz can be found in high proportions in atmospherically transported Saharan dust samples (Avila et al., 1997; Caquineau et al., 1998; Caquineau et al., 2002; Alastuey et al., 2005; Kandler et al., 2009). However, barely any information is available about the alteration of the IN efficiency of quartz due to cloud processing and atmospheric chemical species.

This work and Zolles et al. (2015) have shown a large variability in the IN activity of quartz particles. Milled quartz samples showed high IN activity, while quartz layers grown over 7 months were almost IN inactive. This is in agreement with the

low IN activity of a natural quartz sample investigated by Zolles et al. (2015) with freezing onset < 238 K. The question therefore arises whether naturally eroded quartz particles from atmospherically relevant dust source regions would have significant IN activity.

Kaufmann et al. (2016) performed emulsion freezing experiments with natural dust samples ground-collected in different deserts worldwide and correlated the IN activity with the mineralogical composition. A sample collected in Oman showed

low IN activity despite its considerable quartz content. Conversely, heterogeneous freezing on quartz surfaces may indeed account for the freezing signal observed for higher suspension concentrations of desert dust samples collected in Israel. For the Antarctica sample with 24 % quartz content, Kaufmann et al. (2016) performed emulsion freezing experiments using the sieved fraction and after the sieved fraction was milled. Indeed, the heterogeneously frozen fraction increased due to milling.

Boose et al. (2016b) found a positive correlation between the quartz content and the freezing between 238 K and 245 K of

dust particles sampled from deserts worldwide. However, this positive correlation strongly relies on two milled samples with the highest quartz content (~64 % and ~93 % in samples from Australia and Morocco) which exhibit the highest IN activity. In addition, two more samples were sieved and milled as well. In case of the Atacama sample, with quartz content of ~17 % after sieving and 10 % after milling, milling increased the IN activity. In case of the Israel sample with quartz contents of ~7 % after sieving and ~6 % after milling, the IN activity was decreased after milling. This shows that the IN activity of

collected mineral dusts depends in a complex way on the pre-processing of the samples. More samples need to be collected from desert regions and tested for IN activity without prior milling to assess the correlation between IN activity and quartz content.

## 6 Conclusions and outlook

The analysis of emulsion freezing experiments of quartz and amorphous silica particles allowed to attribute the IN activity of

quartz surfaces to specific surface properties:

     a)   Surface hydroxylation seems to be a necessary but not sufficient condition for the IN activity of quartz and amorphous silica. Silanols may form hydrogen bonds with water molecules that are able to direct them into an ice-like arrangement. However, as it seems, most of the silanols on synthesized amorphous silica and grown quartz surfaces are engaged in hydrogen bonds amongst each other with an insufficient number of OH-groups left for

615          hydrogen bonding to water molecules.

     b)   Milling leads to the cleavage of Si–O–Si bridges resulting in Si–O$^{\bullet}$ and Si$^{\bullet}$ radical sites that react in the presence of water vapor to finally form Si–OH and Si–O–OH on the surface., Defects seem to disrupt the interconnected chains of silanols on the surfaces of milled silica particles, thus increasing the number of silanols that are available for



hydrogen bonding to water molecules. These defects indeed seem to be a prerequisite for the IN activity of amorphous silica and quartz.

c) Usually, amorphous silica particles are synthesized and quartz particles are milled, therefore the crystalline surface of quartz could be expected to template ice. Our emulsion freezing experiments with milled amorphous silica and grown quartz surfaces show that crystallinity is not relevant for the IN activity of silica. Rather a highly defective surface is required.

d) Both, barely IN active grown quartz and highly IN active milled quartz particles carry a negative surface charge at neutral pH conditions. This indicates that surface charge alone is unreliable as a predictor for IN activity.

e) The onset freezing temperatures of quartz suspensions freshly prepared in neutral and acidic salt solutions containing $(NH_4)_2SO_4$, $NH_4HSO_4$, $Na_2SO_4$ follow approximately the prediction of the water-activity-based description, while the heterogeneously frozen fraction increases slightly in the presence of these salts.

f) The IN activity of quartz is decreased in alkaline solutions. The interaction with $NH_3$ deteriorates the IN activity of the quartz surfaces. This is in contrast to the increased IN activity of gibbsite and aluminosilicates i.e. feldspars, mica and kaolinite in the presence of $NH_3$. We ascribe this decrease of IN activity to the increased dissolution of quartz under alkaline conditions. The defects that constitute the active sites seem to be more susceptible and therefore disappear first on a dissolving surface.

g) Suspending quartz particles over days in aqueous solutions at near neutral pH conditions in a glass vial decreases the IN activity considerably. This decrease is reversible and the original IN activity in terms of $F_{het}$ is almost restored when the quartz particles are washed and resuspended in pure water. We assume that a part of the silicic acid leached from the glass vial forms an oligomerized silicic acid layer on top of the quartz surface and blocks the active sites. This layer dissolves when the particles are rinsed in pure water. The formation of this layer might be the first step to quartz growth.

The sensitivity of the IN activity of quartz surfaces to environmental conditions makes it difficult to come to general conclusions regarding the relevance of quartz particles for cloud glaciation. Dry erosion may fracture quartz particles and introduce active sites while wet erosion may destroy active sites. To assess the IN activity of airborne quartz particles, a correlation between the quartz content and the IN activity must be established for samples that did not undergo milling before they were tested for freezing.

### 7 Data availability

The data for freshly prepared quartz suspensions in water or aqueous solutions (Fig. 2) and aging tests (Figs. 5 and 6) presented in this publication are available at the following DOI: 10.3929/ethz-b-000286931.

### Appendix A: Ice nucleation efficiency of quartz from Kaufmann et al. (2016)

A recent study from our group, Kaufmann et al. (2016), reported a very low IN efficiency of a quartz sample suspended in pure water using the same experimental equipment and procedure as in the present study. In emulsion freezing experiments, a heterogeneous freezing signal was observed up to ∼ 247 K, yet with a very low IN active particle fraction (0.01). The quartz sample used in Kaufmann et al. (2016) was procured as a stone from the Institute of Geochemistry and Petrology of ETH Zurich and milled to a fine powder in a tungsten carbide ball mill. Approximately 0.4 wt% of tungsten carbide was introduced as an impurity in the quartz sample due to milling.

To explain the low IN efficiency of the Kaufmann quartz compared with SA quartz, we further characterized these two samples. Scanning electron microscopy (SEM) revealed the presence of giant particles (diameter > 20 µm) in the milled quartz sample from Kaufmann et al. (2016) as shown in Fig. A1, which were absent in SA quartz. SA quartz is dominated by





particles ranging from 0.5 to 10 μm as specified by Sigma-Aldrich. As a hard mineral, quartz is difficult to mill down to very
fine particles without partial amorphization. The amorphous fraction of the Kaufmann quartz was estimated as $6.4 \pm 0.5$ %
based on the background signal observed in the XRD diffractogram compared to $4.5 \pm 0.5$ % for the SA quartz.

The size distribution measurements performed with SMPS/APS in Kaufmann et al. (2016) did not capture the coarse particle
fraction due to inefficient aerosolization of large particles in the fluidized bed. Even when aerosolized, coarse particles can
sediment in the tubing and connections during transport to the SMPS/APS. When calculating the fraction of droplets filled
with particles, the number of particles per sample mass was highly overestimated because the coarse particles did not appear
in the size distribution determined by SMPS/APS. Therefore, in Kaufmann et al. (2016), the homogeneous freezing signal
observed in the emulsion freezing experiments was wrongly assigned to droplets filled with IN inactive quartz particles
although it was mostly due to empty droplets.

To remove the coarse particles present in the Kaufmann quartz, we suspended a concentrated suspension of this sample for
an hour in pure water so that particles with average diameters $\geq 3$ μm should settle. After this time period, the supernatant
was collected and immediately used for an emulsion freezing experiment in order to avoid any further aging of the particles.
The concentration of the supernatant was determined as $\sim 8 - 9$ wt% by evaporating the water and weighing the dried
residual.

In addition, emulsion freezing experiments were carried out on the same supernatant suspension aged for 2 hours, 24 hours
and 72 hours. The DSC thermograms of these measurements are shown in Fig. A2. The heterogeneous freezing onset
temperature of the fresh supernatant was 250.2 K which is even higher than the $T_{het}$ of SA quartz while the heterogeneously
frozen fraction is almost the same for these two quartz samples. This confirms that the low IN efficiency of quartz reported
in Kaufmann et al. (2016) was biased by the presence of giant quartz particles not captured in the SMPS/APS measurements.

**Appendix B: Aging tests on TU Vienna quartz suspended in pure water**

Suspensions of TU Vienna quartz (5 wt%) in pure water were prepared in glass vials and tested over a period of three days.
Immersion freezing experiments were carried out with the DSC setup with emulsions prepared from at least 2 separate
suspensions. Measurements were done on the day of preparation (fresh) and on the subsequent three days in order to assess
the long-term effect of aging on the IN efficiency of the quartz sample. Figure A3 shows the DSC thermograms from
freezing experiments during this aging period. Like SA and Kaufmann quartz, TU Vienna quartz also loses its IN efficiency
drastically over the measured time period. This shows that the decrease in IN efficiency, during aging in conditions with
supersaturated Si concentration, is a common feature of all quartz samples.

*Acknowledgements.* This work was supported by the Swiss National Foundation, project number 200020_156251. We thank
the following colleagues from ETH Zürich: Annette Röthlisberger and Marion Rothaupt for helping in carrying out BET and
XRD measurements and Dr. Michael Plötze for carrying out detailed XRD analysis; Fabian Mahrt for providing the SMPS
and the APS for size distribution measurements; Nadine Borduas and Julie Tolu for their continuous help and support during
ICP-MS measurements; Eszter Barthazy for carrying out SEM analysis. Jonas Fahrni and Dominik Brühwiler from Institute
of Chemistry and Biotechnology (Zürich University of Applied Sciences, Wädenswil (ZHAW)) for the synthesis and
characterization of Stöber particles and functionalization of Alfa Aesar particles. We also like to thank Dr. Alexei Kiselev
from Institute of Meteorology and Climate Research (Karlsruhe Institute of Technology) for providing valuable feedback on
this manuscript.

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





**Table 1** Comparison of IN efficiency (in pure water) of various quartz and amorphous silica particles based on emulsion freezing experiments

| Sample Name | Suspension Concentration (wt %) | $T_{hom}$ (K) | $T_{het}$ (K) | $F_{het}$ |
|---|---|---|---|---|
| Stöber | 10 | 237.0[#] | 237.2* | 0.05 |
| Alfa Aesar | 10 | 237.0[#] | 237.1* | 0.03 |
| Alfa Aesar (Milled) | 10 | 236.8 | 246.8/239.7[@] | 0.42 |
| Quartz (Kaufmann et al., 2016) | 8-9 | 236.5 | 250.2/241.8[@] | 0.70 |
| TU Vienna | 5 | 236.6 | 250.9 | 0.87 |
| Sigma-Aldrich[$] | 5 | 236.5 | 247.6 | 0.82 |
| Sigma-Aldrich[$] | 1 | 236.7 | 246.9 | 0.68 |
| Sigma-Aldrich (Milled)[&] | 5 | 236.4 | 249.0 | 0.92 |
| Sigma-Aldrich (Milled)[&] | 1 | 236.5 | 247.4 | 0.79 |

[#]Mean $T_{hom}$ of pure water emulsions is reported here since the onset of the homogeneous freezing signal cannot be separated from the presumed heterogeneous freezing signal
*Onsets of heterogeneous and homogeneous freezing signals are nearly indistinguishable. The observed onset lies within the precision range of $T_{hom}$ (237.0 K, taken as the point dividing heterogeneous and homogeneous freezing signal to evaluate $F_{het}$)
[@]Onsets of the two shoulders exhibited by these samples (see Fig. 1). $F_{het}$ is based on the whole heterogeneous freezing signal
[$]milled quartz as obtained from Sigma-Aldrich
[&]additionally milled Sigma-Aldrich quartz



**Table 2** Summary of the freezing experiments with emulsified aqueous solution droplets containing Alfa Aesar and Stöber amorphous silica particles (10 wt %). Note that the absolute uncertainty in $F_{het}$ may be up to ± 0.12.

| Sample/Suspension concentration (wt%) | Solute | Solute Concentration (m*/wt%) | $a_w$ | $T_{hom}$ (K)[#] | $T_{het}$ (K) | $F_{het}$ |
|---|---|---|---|---|---|---|
| Alfa Aesar (10 wt %) | $NH_3$ | 0.5m | 0.987 | 236.3 | 236.5 | 0.01 |
| | $NH_3$ | 0.05m | 0.996 | 236.6 | 237.1 | 0.04 |
| | $(NH_4)_2SO_4$ | 1wt% | 0.988 | 236.2 | 236.5 | 0.03 |
| | $(NH_4)_2SO_4$ | 0.05wt% | 0.996 | 236.9 | 237.3 | 0.02 |
| | Pure Water | - | 1 | 237.0 | 237.1 | 0.03 |
| Stöber (10 wt %) | $NH_3$ | 0.5m | 0.997 | 236.3 | 236.2 | 0.01 |
| | $NH_3$ | 0.05m | 0.999 | 236.6 | 236.7 | 0.02 |
| | $(NH_4)_2SO_4$ | 1wt% | 0.988 | 236.2 | 236.3 | 0.02 |
| | $(NH_4)_2SO_4$ | 0.05wt% | 0.994 | 236.9 | 236.9 | 0.02 |
| | Pure Water | - | 1 | 237.0 | 237.2 | 0.05 |

40

*m = molality; [#]Mean $T_{hom}$ of pure water/solution emulsions (taken as the point dividing heterogeneous and homogeneous freezing signal to evaluate $F_{het}$) is reported here since the onset of the homogeneous freezing signal cannot be separated from the presumed heterogeneous freezing signal



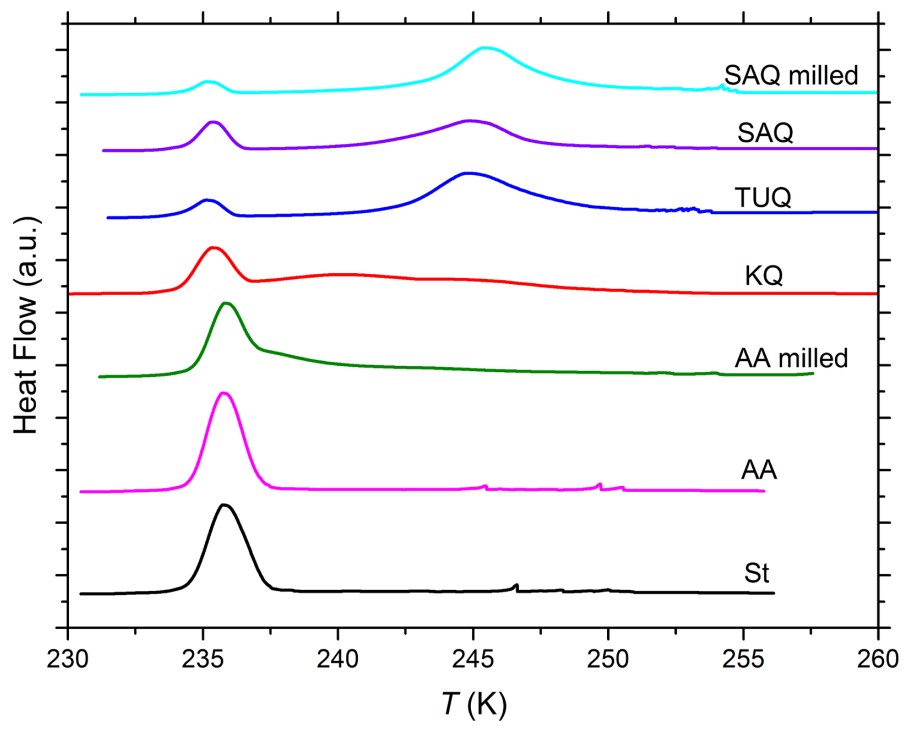

**Figure 1.** DSC thermograms of various quartz and amorphous silica particles suspended in pure water. St: Stöber silica (10 wt%); AA: Alfa Aesar silica (10 wt%); AA milled: milled Alfa Aesar silica (10 wt%); KQ: quartz sample from Kaufmann et al. (2016) (8 – 9 wt%); TUQ: TU Vienna quartz (5 wt%); SAQ: Sigma-Aldrich quartz (5 wt%, milled by Sigma-Aldrich); SAQ milled: additionally milled Sigma-Aldrich quartz (5 wt%). All curves are normalized such that the total areas under the heterogeneous plus homogeneous freezing curves sum up to the same value.





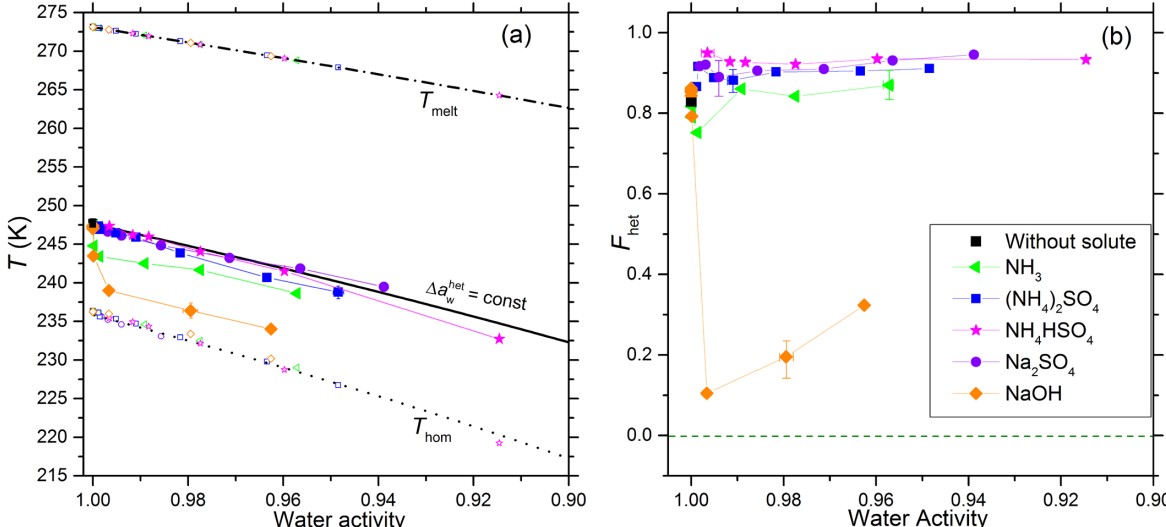

**Figure 2(a).** Onset freezing temperatures of emulsion freezing experiments performed with 5 wt % SA quartz suspended in aqueous solutions of inorganic solutes (for symbols and colors see insert). Heterogeneous freezing onset temperatures, $T_{het}$ (filled solid symbols connected by thin lines), homogeneous freezing onset temperatures, $T_{hom}$ (open symbols at $T < 237$ K), and ice melting temperatures, $T_{melt}$ (open symbols at $T > 262$ K) are given as functions of the solution's water activity, $a_w$. Dash-dotted black line: ice melting curve. Dotted black line: homogeneous ice freezing curve for supercooled aqueous solutions obtained by horizontally shifting the ice melting curve by a constant offset $\Delta a_w^{hom}(T) = 0.30$. Solid black line: horizontally shifted from the ice melting curve by $\Delta a_w^{het}(T) = 0.221$ derived from the heterogeneous freezing temperature of the suspension of quartz in pure water (filled black square at $a_w = 1$). Symbols are the mean of at least two emulsion freezing experiments (using at least two separate suspensions). Two symbols carry error bars to show representative experimental variations (min-to-max) in $T_{het}$ and $a_w$. **(b)** Heterogeneously frozen fraction $F_{het}$ as a function of the solution's water activity ($a_w$). Five symbols carry error bars showing representative experimental variations (min-to-max) in $F_{het}$ and $a_w$. Absolute uncertainties in $F_{het}$ do not exceed $\pm 0.12$.





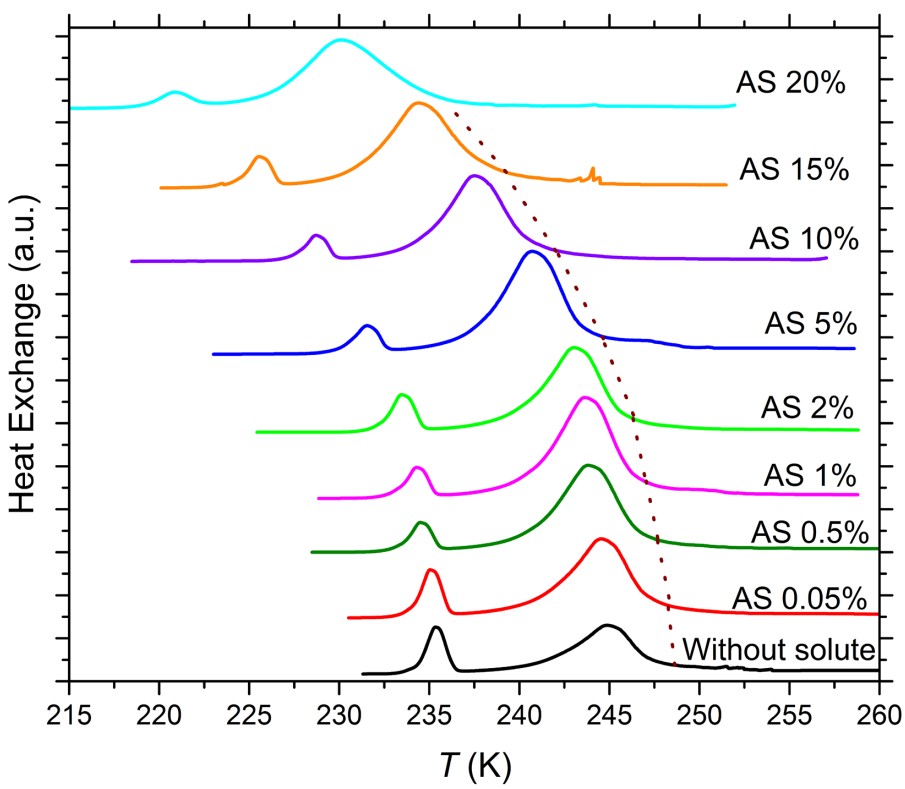

**Figure 3.** DSC thermograms of 5 wt % SA quartz particles suspended in ammonium sulfate (AS) solution droplets of increasing concentrations (0 - 20 wt % AS). All curves are normalized such that the areas under the heterogeneous and homogeneous freezing curves sum up to the same value. The dotted brown line connects the heterogeneous freezing onset temperatures ($T_{het}$) of the emulsions. With increasing AS concentration $T_{het}$ decreases monotonically while the intensity of the heterogeneous freezing signal increases initially in dilute AS solution and remains high up to high solute concentrations.



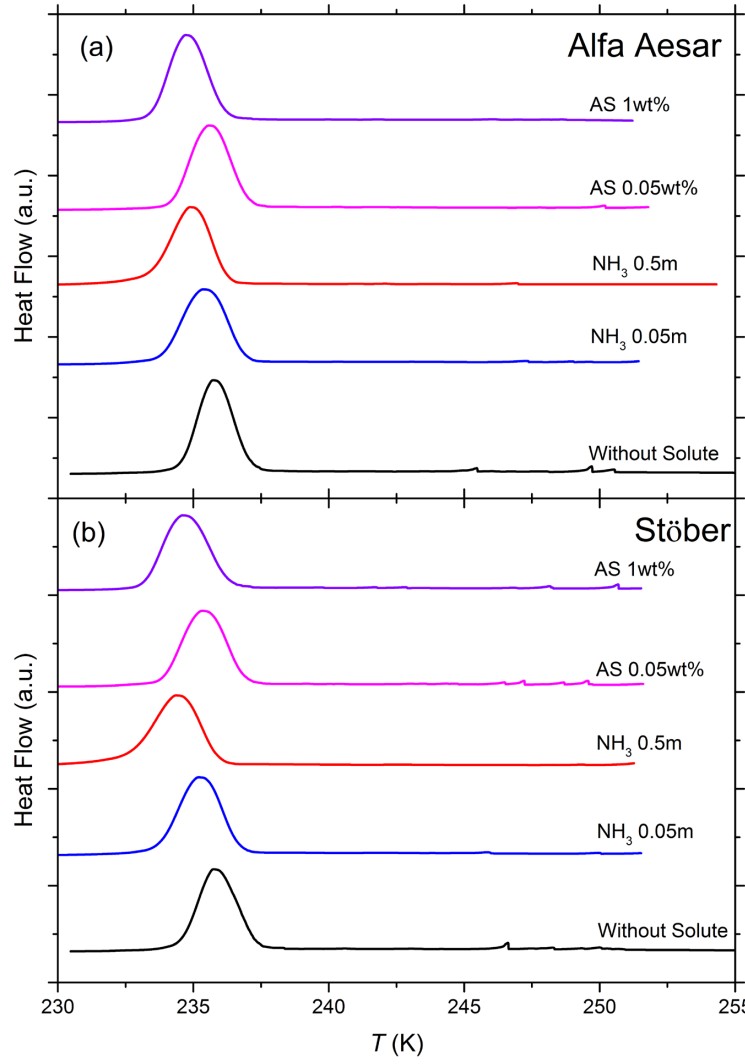


**Figure 4.** DSC thermograms of 10 wt % Alfa Aesar (**panel a**) and 10 wt % Stöber (**panel b**) silica particles suspended in pure water, ammonia ($NH_3$; 0.05 and 0.5 molal) and ammonium sulfate (AS; 0.05 and 1 wt %) solution droplets of varying concentrations (corresponding to $a_w$ range of $1 - 0.987$). All curves are normalized such that the areas under the heterogeneous and homogeneous freezing curves sum up to the same value. DSC thermograms of both amorphous silica samples show only one clear freezing signal which is indistinguishable from the homogeneous freezing signal. $T_{hom}$ of the corresponding emulsion freezing experiments with the pure solutions (in the absence of the silica particles) has been taken as the dividing temperature of heterogeneous and homogeneous freezing to evaluate $F_{het}$ (Table 2; see Section 3.3).





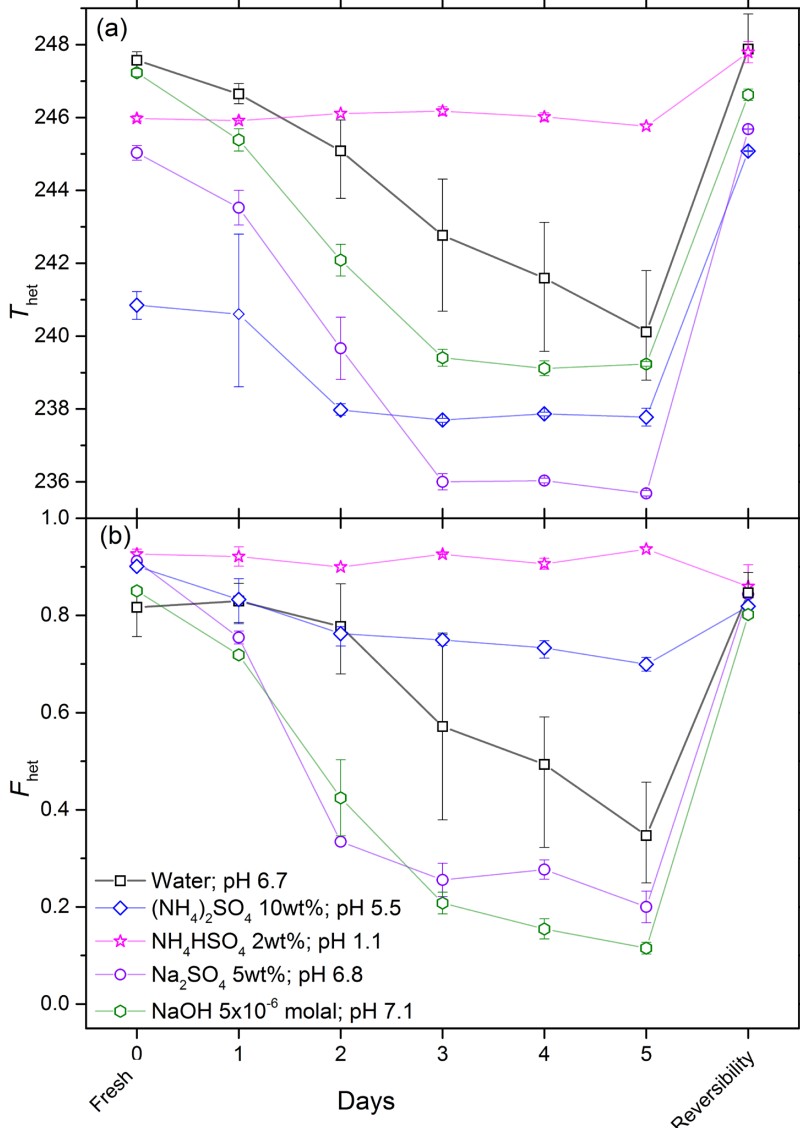

**Figure 5.** Development of $T_{het}$ (**panel a**) and $F_{het}$ (**panel b**) for 5 wt% SA quartz suspended in water, $(NH_4)_2SO_4$ solution (10 wt%), $NH_4HSO_4$ solution (2 wt %), $Na_2SO_4$ solution (5 wt %) and NaOH solution (5 x $10^{-6}$ molal) over a period of five days. All suspensions were prepared and aged in borosilicate glass vials. After five days of aging the reversibility was tested: the suspensions were centrifuged, the supernatant decanted, the aged particles washed several times with pure water, resuspended in pure water, and subjected to an emulsion freezing experiment. All data points are means of at least two separately aged suspensions. The error bars show representative experimental variations (min-to-max).





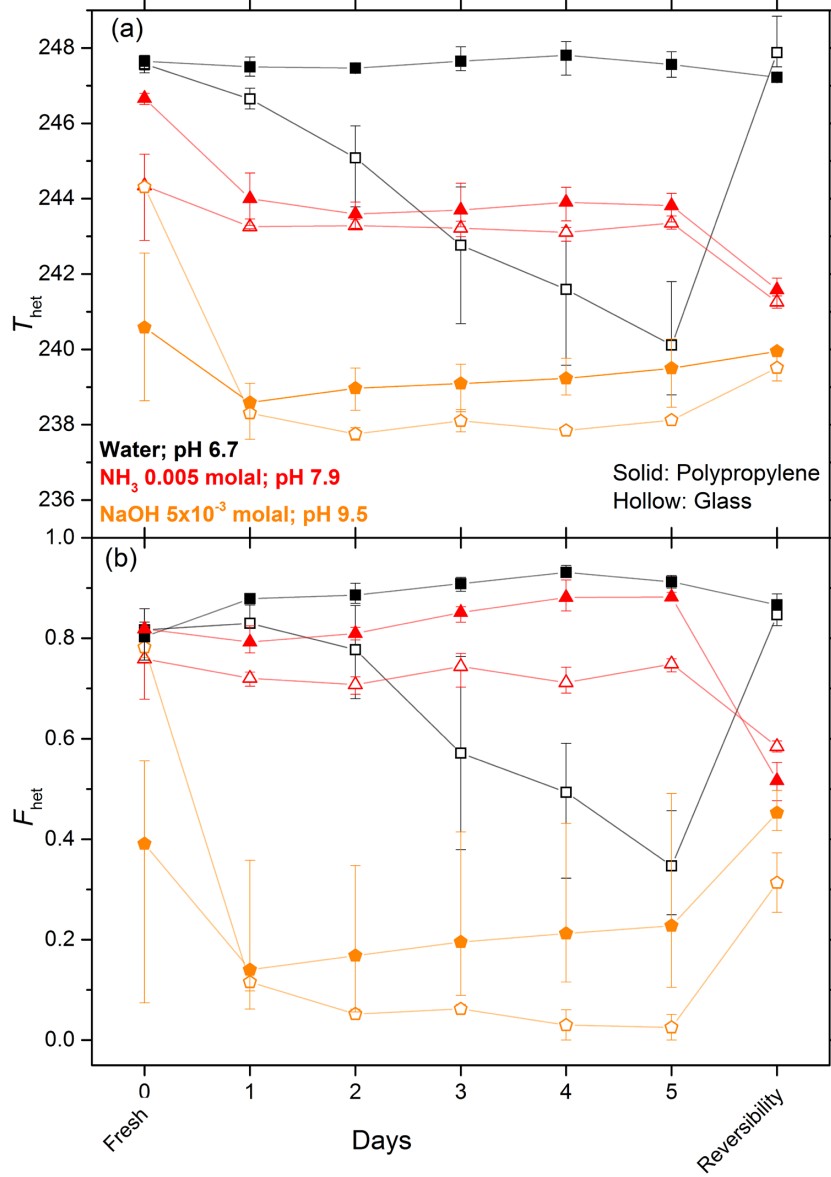


**Figure 6.** Same as Fig. 5, except that it shows a comparison of IN efficiency of SA quartz (5 wt %) suspended in water/aqueous solutions (NH$_3$ 0.005 molal with pH 7.9 and NaOH 5 x 10$^{-3}$ molal with pH 9.5) prepared in borosilicate glass vials (open symbols) and polypropylene falcon tubes (solid symbols). After five days aging the reversibility was tested as explained in Fig. 5.



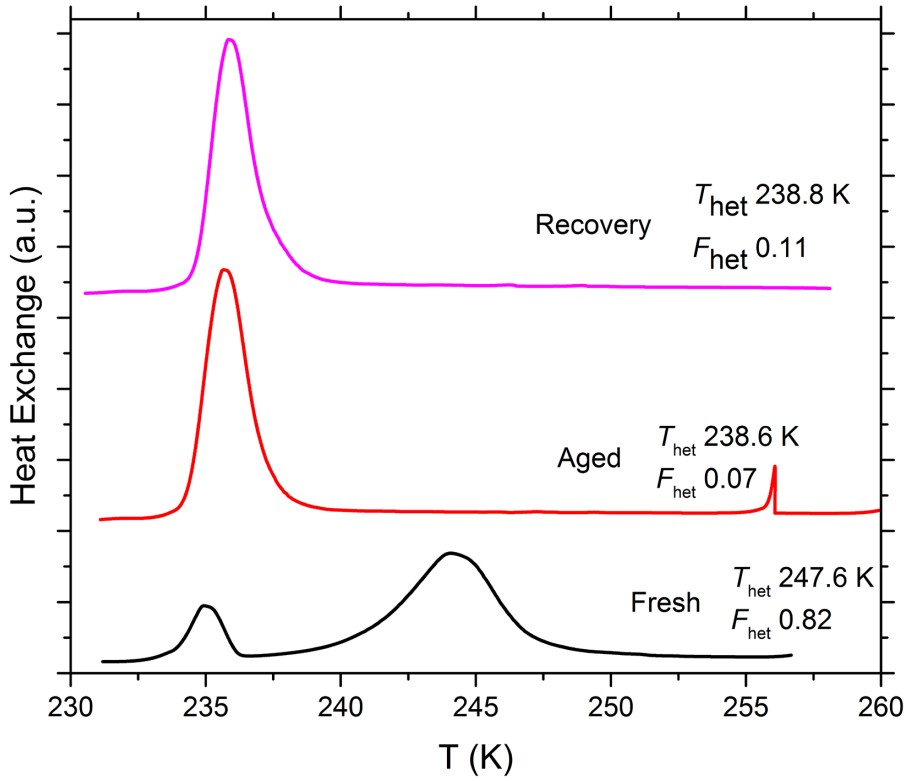


**Figure 7.** DSC thermograms of SA quartz (5 wt %) suspended in pure water in borosilicate glass vials and measured right after preparation (marked as "Fresh"). The same suspension was aged for 7 months and re-measured (marked as "Aged"). The aged suspension was centrifuged, the supernatant decanted, the aged particles washed several times with pure water, resuspended in pure water and measured again to examine the recovery of IN efficiency after aging (marked as "Recovery"). This procedure was
done with 3 different suspensions and their mean $T_{het}$ and $F_{het}$ are reported next to each curve. All curves are normalized such that the areas under the heterogeneous and homogeneous freezing curves sum up to the same value.





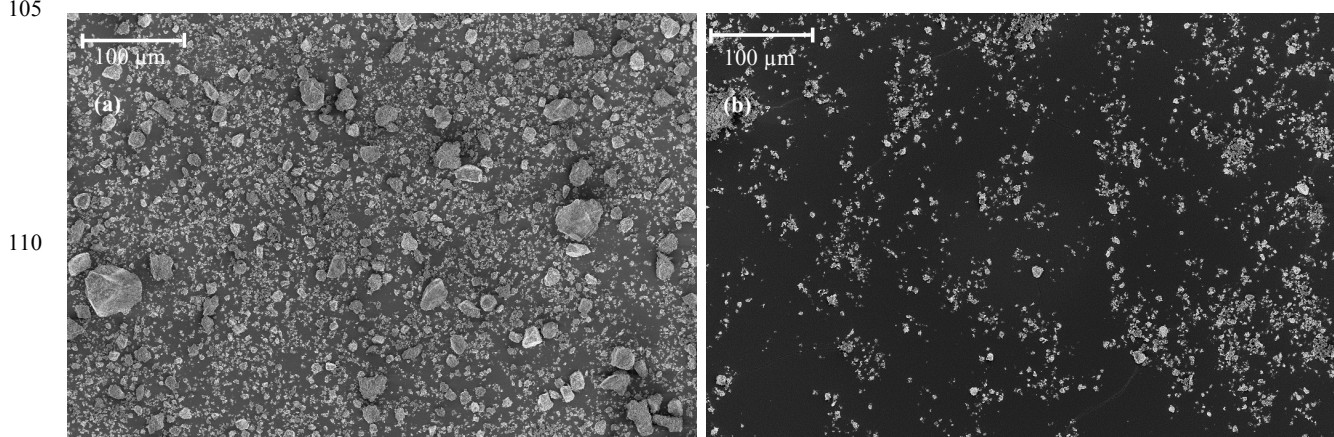

**Figure A1.** SEM images of two different quartz samples at 500X magnification. **(a)** Quartz sample from Kaufmann et al.,
(2016); **(b)** Quartz sample (from Sigma Aldrich) used in this study.



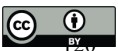

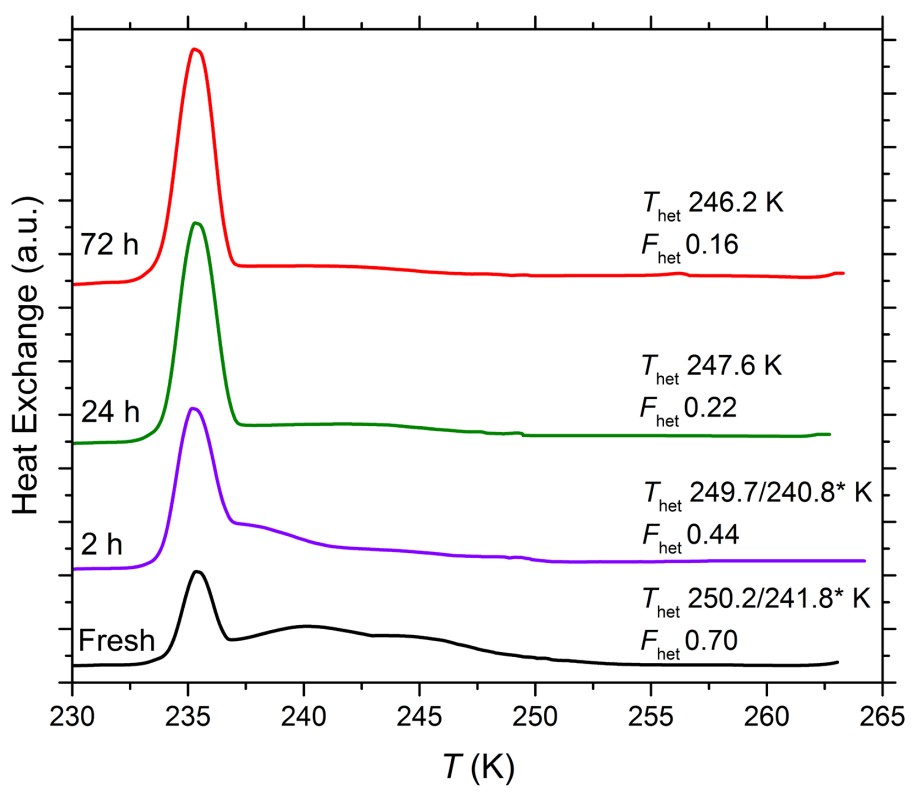

**Figure A2.** DSC thermograms of Kaufmann quartz taken from the supernatant of a suspension of the same quartz sample as used in Kaufmann et al. (2016) that had settled for 1 hour. Settling resulted in particle sizes with diameter $\leq 3$ μm and a suspension concentration of $8 - 9$ wt % in the supernatant. In order to assess the effect of aging, emulsion freezing experiments were performed on the supernatant right after extraction as well as after 2 h, 24 h and 72 h after extraction and the corresponding $T_{het}$ and $F_{het}$ is reported next to each curve. All curves are normalized such that the areas under the heterogeneous and homogeneous freezing curves sum up to the same value. The (*) represents the onset temperatures of the two shoulders of the heterogeneous freezing signal and the corresponding $F_{het}$ is calculated based on the complete freezing signal.





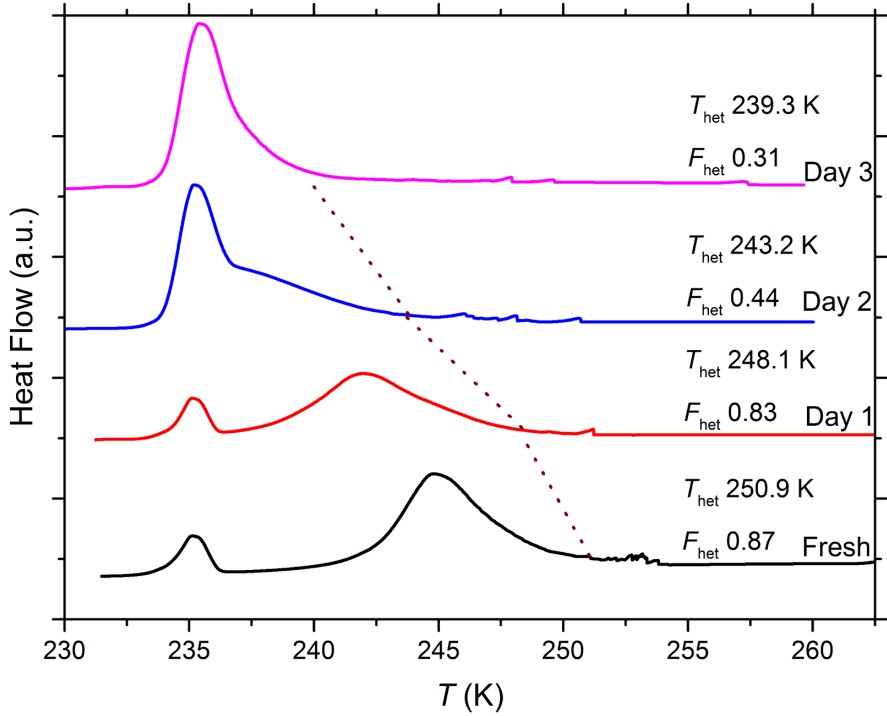


**Figure A3.** DSC thermograms of 5 wt % TU Vienna quartz suspension in pure water, obtained via emulsion freezing experiments performed on the day of suspension preparation (fresh) and the subsequent three days and the corresponding $T_{het}$ and $F_{het}$ is reported next to each curve. Suspensions were prepared in borosilicate glass vials. All curves are normalized such that the areas under the heterogeneous and homogeneous freezing curves sum up to the same value.