# Peer review of "Ice nucleation activity of silicates and aluminosilicates in pure water and aqueous solutions. Part 2 — Quartz and amorphous silica"

_Atmospheric Chemistry and Physics, 2018_

## Referee Comment (RC1) · Anonymous Referee #1 · 8 Jan 2019

General comments

This paper examines immersion mode ice nucleation by a range of quartz and silica samples using differential scanning calorimeter technique which allows comparison of different nucleators and solute conditions, concluding that milling processes produce the sites that cause quartz to nucleate ice and that quartz doesn't nucleate ice better in the presence of ammonia, in contrast to feldspar minerals. The paper provides numerous sensible and well-supported suggestions for factors that may affect the effectiveness of quartz and silica and ice nucleators and discusses the complexity of assessing the role of quartz in atmospheric ice nucleation. The paper is admirably

thorough and is well written. The conclusions are within in the scope of ACP and are of substantial interest. While I have a few minor comments that the authors may want to consider I support publication.

Specific comments

I think it is generally agreed that calculation of ice nucleation active site densities is helpful e.g. (Hoose and Möhler, 2012) and a recent intercomparison suggests that comparisons across different instruments is likely meaningful (DeMott et al., 2018). The data produced here and in other papers using DSC provides useful information for internal comparison but it is rather a shame that it does not lend itself to quantitative comparison with other techniques. Would it be possible to calculate ns values from the available data? Some comment on this might be helpful.

Relatedly, the weight percents of quartz used are very high. A 10 wt% suspension is almost like mud and will flocculate and settle effectively immediately. I am curious as to why such high particle concentrations have been used? In addition, the presence of solutes may conceivable change the rate of flocculation and change how the solutions interact with the emulsifying oil. For the comparisons made in this study to be valid the droplets it is, I think, necessary that the droplet distribution across all DSC experiments is very similar. While I recognise that the original paper on the technique paper used up to 20 wt% I still think it would be helpful to have a figure or table somewhere demonstrating that droplet distributions for different suspension compositions are indeed similar, and perhaps some brief comment on how any variation in this distribution may affect results.

The paper is quite long and much of the midsection is a rather turgid. Obviously, this paper is just one part of a very substantial of work, indicating sensible efforts have been made to divide up the vast number of results. However, the authors may want to consider presenting their results more concisely in places. This would aid readability.

Pg 9 line 334-336- The surface area per droplet in Whale et al. is not as large as

Interactive
comment
10 cmˆ2. The droplets used were smaller than those of Zobrist with a similar particle concentration. I suspect the silica used was simply different in some and that this accounts for the different freezing temperature observed.

Technical comments

Pg 3 line 77 and other places- There is some inconsistency in the citations to the other papers in the Kumar series, a's and b's aren't always present.

Pg 3 line 105- Word missing somewhere in the sentence starting 'Quartz,....'

Pg 4 line 127- remove 'only a mere...'

Pg 4 line 138- different number of significant figures on weight loss compared to else-where.

Pg 10 line 351-354- This sentence is a little difficult to follow.

Pg 11 line 413- 'Shortly' is the wrong word here, 'briefly' maybe, although it would be better to just state the length of time.

Pg 12 line 436- 'little' is the wrong word. Should be 'few'

Pg 13 line 477- I'm not clear what is meant by 'when quartz is ground in the atmo-sphere'. This should probably be explained.

Pg 17 line 630- I'm not sure 'deteriorates' is the right word.

References

DeMott, P. J., Möhler, O., Cziczo, D. J., Hiranuma, N., Petters, M. D., Petters, S. S., Belosi, F., Bingemer, H. G., Brooks, S. D., Budke, C., Burkert-Kohn, M., Collier, K. N., Danielczok, A., Eppers, O., Felgitsch, L., Garimella, S., Grothe, H., Herenz, P., Hill, T. C. J., Höhler, K., Kanji, Z. A., Kiselev, A., Koop, T., Kristensen, T. B., Krüger, K., Kulka-rni, G., Levin, E. J. T., Murray, B. J., Nicosia, A., O'Sullivan, D., Peckhaus, A., Polen, M. J., Price, H. C., Reicher, N., Rothenberg, D. A., Rudich, Y., Santachiara, G., Schiebel,

T., Schrod, J., Seifried, T. M., Stratmann, F., Sullivan, R. C., Suski, K. J., Szakáll, M., Taylor, H. P., Ullrich, R., Vergara-Temprado, J., Wagner, R., Whale, T. F., Weber, D., Welti, A., Wilson, T. W., Wolf, M. J., and Zenker, J.: The Fifth International Workshop on Ice Nucleation phase 2 (FIN-02): laboratory intercomparison of ice nucleation measurements, Atmos. Meas. Tech., 11, 6231-6257, 10.5194/amt-11-6231-2018, 2018.

Hoose, C., and Möhler, O.: Heterogeneous ice nucleation on atmospheric aerosols: a review of results from laboratory experiments, Atmos. Chem. Phys., 12, 9817-9854, 10.5194/acp-12-9817-2012, 2012.

---

## Referee Comment (RC2) · Anonymous Referee #2 · 19 Jan 2019

The manuscript I was asked to evaluate is the second part of the publication series dedicated to the experimental study of freezing behavior of several types of mineral dust particles immersed in the droplets of pure water and weak inorganic solutions. The motivation behind this work is not only the persisting need to understand the heterogeneous nucleation of ice on the molecular level, but also a demand to establish a model framework that would incorporate the original ice nucleating (IN) properties of aerosol particles, their chemical ageing under atmospheric conditions, and interaction with water and water vapor in tropospheric clouds. This complete paper series and the part 2 in particular is undoubtedly one of the most comprehensive experimental studies of various effects arising from the interaction of water, solute, and mineral surfaces

that have been published recently. While I fully support the publication, I do have a few critical remarks that the authors might want to address while preparing the final version of the manuscript.

Abstract, lines 10-11: "We performed immersion freezing experiments and relate the reported contradictory behavior to the influence of milling, and to the aging time and conditions since milling." It is not clear to me what contradictory behavior is meant here. Is that some behavior contradictory to what has been published previously or any inconsistency within your own results? In any case, this is too much information for one sentence in an abstract. Please reformulate the abstract in a more concise way.

The milling as a factor controlling ice nucleating properties of quartz is mentioned already in the second sentence of the abstract and then discussed throughout the text. It creates an impression that effect of milling on IN activity of quartz is the main objective of the study, and that all quartz samples have been milled by the authors. Only later it becomes clear that all samples have been milled by the manufacturer and thus any commercially available samples of crystalline quartz are called "milled". Perhaps this should be made clear at the beginning.

Lines 173-175. Why the freezing and the melting temperatures on the DSC thermograms are treated in a different way: as a leading edge for freezing peak, and peak maximum for melting? How narrow are melting peaks, would that make any difference if the melting temperature would be measured at the leading edge? Could you include a typical melting thermograms into the supporting material?

I don't quite understand the need for SMPS/APS measurements of aero-dispersed samples since they were not used for any DSC studies. The dispersion method always introduces some distortion into the initial particle size distribution in the powder, due to the intrinsic size selectivity. Large particles don't make it around the bends of the tubing, the small ones would be lost via diffusion deposition, there are electrostatic charging effects etc. Is there any physical reason for bimodal size distribution?

What density and aerodynamic shape factor have been used for calculation of volume-equivalent diameter of SMPS and APS data? I have a feeling that the SMPS/DMA data is an unnecessary piece of information that can be easily omitted from the manuscript.

Finally, I am curios if the PZC-based discussion of surface charge role for the IN properties of quartz (section 4.5.4) makes much sense given the variability of the hydroxyl group number density across different crystalline faces of quartz (as you state yourself in lines 502 - 503). Since quartz does not have cleavage planes, the PZC measured with the conventional method is an effective value that does not reflect local anomalies. For ice nucleation, any surface patch larger than the size of a critical nucleus that would have the right surface charge would trigger the nucleation of ice. I, therefore, don't see any contradiction between your results and the work of Abdelmonem, (2017) and suggest that you remove the generalizing sentence (line 550) from the manuscript.

Abdelmonem, A., et al. (2017). Surface-Charge-Induced Orientation of Interfacial Water Suppresses Heterogeneous Ice Nucleation on $\alpha$-Alumina (0001), Atmos. Chem. Phys. 17(12): 7827.

---

## Author Comment (AC1) · 25 Mar 2019

We thank Reviewer 1 for his/her constructive comments. We reproduce reviewer's comments in *blue* and our responses in black.

*General comments:*
*This paper examines immersion mode ice nucleation by a range of quartz and silica samples using differential scanning calorimeter technique which allows comparison of different nucleators and solute conditions, concluding that milling processes produce the sites that cause quartz to nucleate ice and that quartz doesn't nucleate ice better in the presence of ammonia, in contrast to feldspar minerals. The paper provides numerous sensible and well-supported suggestions for factors that may affect the effectiveness of quartz and silica and ice nucleators and discusses the complexity of assessing the role of quartz in atmospheric ice nucleation. The paper is admirably thorough and is well written. The conclusions are within in the scope of ACP and are of substantial interest. While I have a few minor comments that the authors may want to consider I support publication.*

*Specific Comments:*
*I think it is generally agreed that calculation of ice nucleation active site densities is helpful e.g. (Hoose and Möhler, 2012) and a recent intercomparison suggests that comparisons across different instruments is likely meaningful (DeMott et al., 2018). The data produced here and in other papers using DSC provides useful information for internal comparison but it is rather a shame that it does not lend itself to quantitative comparison with other techniques. Would it be possible to calculate ns values from the available data? Some comment on this might be helpful.*

Indeed, it is a drawback of the DSC method that ice nucleation active site densities cannot be directly derived. We attempted to quantify the ice nucleation activity in Kaufmann et al. (2016), by deriving an ice active particle fraction ($f_{act}$) from emulsion freezing experiments. However, there are significant uncertainties associated with the dust particle size distribution, the droplet size distribution and frozen water volume fractions in cases with overlapping heterogeneous and homogeneous freezing signals, resulting in $f_{act}$ that might be up to a factor 6 higher or more than a factor 2 lower. An ice nucleation active site density as a function of temperature would still be more uncertain because the DSC does not register freezing events but just the heat flow due to freezing, which is shifted to lower temperatures compared with the freezing events. For TU Vienna quartz and SA quartz that we investigated in this study, active site densities have already been given in Zolles et al. (2015) and Whale et al. (2018), respectively. The focus of this study was to investigate how solutes affect the ice nucleation activity. DSC emulsion freezing measurements are very well suited for this purpose because they yield good statistics within one experiment. Adding active site densities would need an in-depth discussion of how they were derived from the DSC thermograms. Since active site densities of quartz samples investigated in this study have been reported previously and the manuscript is long already, we would like to refrain from adding this information.

*Relatedly, the weight percents of quartz used are very high. A 10 wt% suspension is almost like mud and will flocculate and settle effectively immediately. I am curious as to why such high particle concentrations have been used?*

As a standard, we used 5 wt% suspension concentrations. This concentration exhibits a low viscosity and can be well handled. We increased the suspension concentrations to 10 wt% in cases when 5 wt% did not lead to a discernable heterogeneous freezing signal. This was the case for the original amorphous silica sample (Alfa Aesar). For better comparison, we decided to also present the milled Alfa Aesar sample with a 10 wt%

suspension concentration. Moreover, all suspensions were strongly sonicated for 5 min before preparing the emulsions to minimize aggregation.

*In addition, the presence of solutes may conceivable change the rate of flocculation and change how the solutions interact with the emulsifying oil. For the comparisons made in this study to be valid the droplets it is, I think, necessary that the droplet distribution across all DSC experiments is very similar. While I recognize that the original paper on the technique paper used up to 20 wt% I still think it would be helpful to have a figure or table somewhere demonstrating that droplet distributions for different suspension compositions are indeed similar, and perhaps some brief comment on how any variation in this distribution may affect results.*

In Marcolli et al. (2007) we investigated whether the added dust particles influence the droplet size distribution. For Arizona test dust, we did not find a significant dependence up to 20 wt% suspension concentration. We therefore used all analyzed suspension concentrations to obtain a representative droplet size distribution. This is stated in Marcolli et al. (2007).

We regularly inspect the emulsions under the microscope to make sure that they are of uniform quality. We also did this after having added solutes and did not find any effect of the solutes on the droplet size distribution. Moreover, strong interactions of the added particles or solutes with the surfactant would become visible in the DSC thermograms as large spikes at warmer temperature due to large patches of water present in the emulsion that freeze at warmer temperature. We have added a sentence to the methodology section (lines 163-164) to make clear that we regularly monitor the quality of the emulsions: "Regular inspection under the microscope did not reveal an effect of dust particles or solutes on the droplet size distribution."

*The paper is quite long and much of the midsection is a rather turgid. Obviously, this paper is just one part of a very substantial of work, indicating sensible efforts have been made to divide up the vast number of results. However, the authors may want to consider presenting their results more concisely in places. This would aid readability.*

We appreciate the reviewer's comment and have made appropriate changes in the revised manuscript to be more concise in the *Discussion* section. Lines 326 – 330, 454 – 456 and 528 - 532 (from the 1st submission) have been deleted.

*Pg 9 line 334-336- The surface area per droplet in Whale et al. is not as large as 10 cm^2. The droplets used were smaller than those of Zobrist with a similar particle concentration. I suspect the silica used was simply different in some and that this accounts for the different freezing temperature observed.*

Indeed, our formulation was imprecise, we therefore re-phrase: "Whale et al. (2018) found IN active site densities of silica particles from Sigma-Aldrich (silica gel) of $n_s \approx 10$ cm$^{-2}$ at 251 K and $n_s \approx 0.1$ cm$^{-2}$ at 261 K, corresponding to a slightly higher IN activity of these silica particles compared with those synthesized by Zobrist et al. (2008)." in the revised manuscript (lines 331 – 333).

*Pg 3 line 77 and other places- There is some inconsistency in the citations to the other papers in the Kumar series, a's and b's aren't always present.*

All inconsistencies have been corrected in the revised manuscript (line 187 and 537).

*Pg 3 line 105- Word missing somewhere in the sentence starting 'Quartz,. . ..'*

We corrected by adding: "Quartz, the most common form of crystalline silica, …" in the revised manuscript (line 105).

*Pg 4 line 127- remove 'only a mere. . .'*
We removed it.

*Pg 4 line 138- different number of significant figures on weight loss compared to elsewhere.*
Weight loss data has been adjusted to 2 significant digits (line 138).

*Pg 10 line 351-354- This sentence is a little difficult to follow.*
We split this sentence in the revised manuscript (lines 348 - 351): "In Part 1 of this series, we showed that heterogeneous freezing onsets of microcline exhibit strong deviations from the water-activity-based description. Higher $T_{het}$ compared to predicted values were observed for microcline suspended in very dilute $NH_3/NH_4^+$ containing solutions, while a substantial decrease in IN efficiency was observed in more concentrated solutions of inorganic salts including $NH_4^+$ containing salts and $NH_3$."

*Pg 11 line 413- 'Shortly' is the wrong word here, 'briefly' maybe, although it would be better to just state the length of time.*
We revised by just stating the length of time: "for the sample exposed to water for ~ 15 min" in the revised manuscript (lines 413 - 414).

*Pg 12 line 436- 'little' is the wrong word. Should be 'few'*
We replaced "little" by "few" (line 436).

*Pg 13 line 477- I'm not clear what is meant by 'when quartz is ground in the atmosphere'. This should probably be explained.*
We mean here "ground in the normal atmosphere" compared to grinding in a protected atmosphere devoid of oxygen. We therefore revised to "when quartz is ground in the normal atmosphere" (line 473).

*Pg 17 line 630- I'm not sure 'deteriorates' is the right word.*
We replaced "deteriorates" by "suppresses" (line 621).

**References**

Kaufmann, L., Marcolli, C., Hofer, J., Pinti, V., Hoyle, C. R., and Peter, T.: Ice nucleation efficiency of natural dust samples in the immersion mode, Atmos. Chem. Phys., 16, 11177-11206, doi:10.5194/acp-16-11177-2016, 2016.

Marcolli, C., Gedamke, S., Peter, T., and Zobrist, B.: Efficiency of immersion mode ice nucleation on surrogates of mineral dust, Atmos. Chem. Phys., 7, 5081-5091, doi:10.5194/acp-7-5081-2007, 2007.

Whale, T. F., Holden, M. A., Wilson, Theodore W., O'Sullivan, D., and Murray, B. J.: The enhancement and suppression of immersion mode heterogeneous ice-nucleation by solutes, Chemical Science, 9, 4142-4151, doi:10.1039/C7SC05421A, 2018.

Zobrist, B., Marcolli, C., Peter, T., and Koop, T.: Heterogeneous ice nucleation in aqueous solutions: The role of water activity, The Journal of Physical Chemistry A, 112, 3965-3975, doi:10.1021/jp7112208, 2008.

Zolles, T., Burkart, J., Häusler, T., Pummer, B., Hitzenberger, R., and Grothe, H.: Identification of ice nucleation active sites on feldspar dust particles, The Journal of Physical Chemistry A, 119, 2692-2700, doi:10.1021/jp509839x, 2015.

---

## Author Comment (AC2) · 25 Mar 2019

We thank Reviewer 2 for his/her constructive comments. We reproduce reviewer's comments in *blue* and our responses in black.

*The manuscript I was asked to evaluate is the second part of the publication series dedicated to the experimental study of freezing behavior of several types of mineral dust particles immersed in the droplets of pure water and weak inorganic solutions. The motivation behind this work is not only the persisting need to understand the heterogeneous nucleation of ice on the molecular level, but also a demand to establish a model framework that would incorporate the original ice nucleating (IN) properties of aerosol particles, their chemical ageing under atmospheric conditions, and interaction*
*with water and water vapor in tropospheric clouds. This complete paper series and the part 2 in particular is undoubtedly one of the most comprehensive experimental studies of various effects arising from the interaction of water, solute, and mineral surfaces that have been published recently. While I fully support the publication, I do have a few critical remarks that the authors might want to address while preparing the final version of the manuscript.*

*Abstract, lines 10-11: "We performed immersion freezing experiments and relate the reported contradictory behavior to the influence of milling, and to the aging time and conditions since milling." It is not clear to me what contradictory behavior is meant here. Is that some behavior contradictory to what has been published previously or any inconsistency within your own results? In any case, this is too much information for one sentence in an abstract. Please reformulate the abstract in a more concise way.*

We agree that the meaning of "*contradictory behavior*" is not clear. We therefore re-formulate: "We performed immersion freezing experiments and relate the observed variability in IN activity to the influence of milling, the aging time and to the exposure conditions since milling." (lines 12 – 13)

*The milling as a factor controlling ice nucleating properties of quartz is mentioned already in the second sentence of the abstract and then discussed throughout the text. It creates an impression that effect of milling on IN activity of quartz is the main objective of the study, and that all quartz samples have been milled by the authors. Only later it becomes clear that all samples have been milled by the manufacturer and thus any commercially available samples of crystalline quartz are called "milled". Perhaps this should be made clear at the beginning.*

This is a valid point. We add a sentence to the abstract to make clear that most quartz samples are obtained by milling (page 1, lines 8 – 9): "Since most studies so far reported IN activities of commercial quartz dusts that were milled already by the manufacturer, IN active samples prevailed."

*Lines 173-175. Why the freezing and the melting temperatures on the DSC thermograms are treated in a different way: as a leading edge for freezing peak, and peak maximum for melting? How narrow are melting peaks, would that make any difference if the melting temperature would be measured at the leading edge? Could you include a typical melting thermograms into the supporting material?*

We chose the heterogeneous freezing onset to characterize the freezing temperature because it is a very well defined parameter easily evaluated from the thermograms. Moreover, the maxima of heterogeneous freezing is not detectable in cases where the heterogeneous freezing peak is only a shoulder on the homogeneous freezing peak. Moreover, the maximum observed in the DSC thermograms is not the maximum of freezing events, but the maximum of heat flow. Therefore, we consider the freezing onset as the more characteristic

quantity for evaluating heterogeneous freezing. Furthermore, reporting the onset is consistent with previous studies on homogeneous ice nucleation such as Koop et al. (2000).

While melting of ice occurs at one temperature, ice within a freeze concentrated solution shows eutectic melting yielding a melting peak with no well-defined onset. Since ice remains in thermodynamic equilibrium with the freeze concentrated solution, it melts gradually, such that the melting signal constantly increases until it peaks when ice completely melts. Therefore, the peak of the melting curve can be used to evaluate the water activity of the solution.

We added melting thermograms for pure water and 5 wt% $(NH_4)_2SO_4$ to the Supplementary Material (Figure S30).

*I don't quite understand the need for SMPS/APS measurements of aero-dispersed samples since they were not used for any DSC studies. The dispersion method always introduces some distortion into the initial particle size distribution in the powder, due to the intrinsic size selectivity. Large particles don't make it around the bends of the tubing, the small ones would be lost via diffusion deposition, there are electrostatic charging effects etc. Is there any physical reason for bimodal size distribution?*

The SMPS/APS measurements were carried out to obtain a rough estimate of the particle size distribution and to verify the size distribution information provided by the manufacturer. In addition, scanning electron microscopy was conducted on quartz samples. It ensured the presence of enough micrometer or smaller size particles to fill most water/solution droplets with at least one particle. We do not know the reason for the bimodal size distribution of the quartz sample. It might depend on the sample-handling procedure adopted by the manufacturer.

*What density and aerodynamic shape factor have been used for calculation of volume equivalent diameter of SMPS and APS data? I have a feeling that the SMPS/DMA data is an unnecessary piece of information that can be easily omitted from the manuscript.*

Density 2.6 g cm$^{-3}$ and shape factor of 1 have been used for the calculation of volume equivalent diameter from SMPS and APS data. The coarseness of the sample needs to be roughly known since it influences the fraction of droplets in the emulsion that contain at least one dust particle.

*Finally, I am curios if the PZC-based discussion of surface charge role for the IN properties of quartz (section 4.5.4) makes much sense given the variability of the hydroxyl group number density across different crystalline faces of quartz (as you state yourself in lines 502 - 503). Since quartz does not have cleavage planes, the PZC measured with the conventional method is an effective value that does not reflect local anomalies. For ice nucleation, any surface patch larger than the size of a critical nucleus that would have the right surface charge would trigger the nucleation of ice. I, therefore, don't see any contradiction between your results and the work of Abdelmonem, (2017) and suggest that you remove the generalizing sentence (line 550) from the manuscript.*

This is a good point. We are also skeptical whether there is a general effect of surface charge or whether the effect of surface charge depends on the specific mineral surface. We therefore remove the generalizing sentence as suggested by the reviewer.

**References**

Koop, T., Luo, B., Tsias, A., and Peter, T.: Water activity as the determinant for homogeneous ice nucleation in aqueous solutions, Nature, 406, 611-614, 2000.

---

## Author Response (AR2)

We give below our responses (in black) to Co-Editor's comments (*in blue*)

*Dear Authors,*

*Thank you for submitting your revised manuscript that has addressed the referees' comments and questions. I am happy to accept your work for publication in ACP.*

We thank the co-Editor for the appreciation and constructive comments.

*In the Introduction, and elsewhere, there are some other well-cited studies of the effects of chemical aging on the ice nucleation properties of mineral dust that you might consider including and discussing briefly. The FROST-2 campaign at LACIS condensed sulphuric acid onto mineral aerosol particles, and found this inhibited the deposition freezing, while immersion freezing was partially decreased.*

*Sullivan, R. C.; Petters, M. D.; DeMott, P. J.; Kreidenweis, S. M.; Wex, H.; Niedermeier, D.; Hartmann, S.; Clauss, T.; Stratmann, F.; Reitz, P.; Schneider, J.; Sierau, B. Irreversible loss of ice nucleation active sites in mineral dust particles caused by sulphuric acid condensation. Atmos. Chem. Phys. 2010, 10, 11471–11487.*

*Reitz, P.; Spindler, C.; Mentel, T. F.; Poulain, L.; Wex, H.; Mildenberger, K.; Niedermeier, D.; Hartmann, S.; Clauss, T.; Stratmann, F.; Sullivan, R. C.; DeMott, P. J.; Petters, M. D.; Sierau, B.; Schneider, J. Surface modification of mineral dust particles by sulphuric acid processing: implications for ice nucleation abilities. Atmos. Chem. Phys. 2011, 11, 7839–7858, doi:10.5194/acp-11-7839-2011.*

*Niedermeier, D.; Hartmann, S.; Wex, H.; Clauss, T.; Kiselev, A.; Sullivan, R. C.; DeMott, P. J.; Petters, M. D.; Reitz, P.; Schneider, J.; Mikhailov, E.; Reimann, B.; Bundke, U.; Stetzer, O.; Sierau, B.; Shaw, R. A.; Mentel, T. F.; Stratmann, F. Experimental study of the role of physicochemical surface processing on the IN ability of mineral dust particles. Atmos. Chem. Phys. 2011, 11, 11131–11144.*

The suggested references have been added to the revised manuscript (lines 77 – 78).

*In related experiments, Arizona test dust (which Atkinson et al. later found contains K-feldspars, which explains the high INA of this dust) was exposed to nitric acid vapor. This was found to impair deposition freezing while immersion freezing was unaltered. The possible reasons for the differences between the effects of sulphuric and nitric acid were discussed. It was concluded that the nitric acid reaction products dissolve off the surface in the immersion mode to either re-expose the original ice active sites, or new surface sites if irreversible chemical alteration of the surface resulted.*

*Sullivan, R. C.; Miñambres, L.; DeMott, P. J.; Prenni, A. J.; Carrico, C. M.; Levin, E. J. T.; Kreidenweis, S. M. Chemical processing does not always impair heterogeneous ice nucleation of mineral dust particles. Geophys. Res. Lett. 2010, 37, doi:10.1029/2010GL045540.*

*The role of the reaction product versus chemical alteration of surface sites was then investigated by Sihvonen et al.:*

*Sihvonen, S. K.; Schill, G. P.; Lyktey, N. A.; Veghte, D. P.; Tolbert, M. A.; Freedman, M. A. Chemical and physical transformations of aluminosilicate clay minerals due to Acid treatment and consequences for heterogeneous ice nucleation. J. Phys. Chem. A 2014, 118, 8787–96, doi:10.1021/jp504846g.*

The suggested reference has been added to the revised manuscript (lines 78 – 80).

*Finally, just recently a study on the effect of secondary organic aerosol coatings on the INA of mineral dust concluded that there was no discernible effect in the immersion freezing mode. Similar to the nitric acid aging, presumably the SOA is dissolved off the surface when immersed in droplets to uncover the surface sites. This paper was just recently accepted for publication in ACP:*

*Heterogeneous Ice Nucleation Properties of Natural Desert Dust Particles Coated with a Surrogate of Secondary Organic Aerosol*

*Zamin A. Kanji, Ryan C. Sullivan, Monika Niemand, Paul J. DeMott, Anthony J. Prenni, Cedric Chou, Harald Saathoff, and Ottmar Möhler*

*Atmos. Chem. Phys. Discuss., https://doi.org/10.5194/acp-2018-905, 2018*

*Revised manuscript accepted for ACP (discussion: closed, 4 comments)*

The suggested reference has been added to the revised manuscript (lines 80 – 82).

[revised manuscript text omitted]